# Asymmetric magnetization switching and programmable complete Boolean logic enabled by long-range intralayer Dzyaloshinskii-Moriya interaction

Qianbiao Liu[1,2], Long Liu [3,5], Guozhong Xing [3,4] & Lijun Zhu [1,2] ✉

After decades of efforts, some fundamental physics for electrical switching of magnetization is still missing. Here, we report the discovery of the long-range intralayer Dzyaloshinskii-Moriya interaction (DMI) effect, which is the chiral coupling of orthogonal magnetic domains within the same magnetic layer via the mediation of an adjacent heavy metal layer. The effective magnetic field of the long-range intralayer DMI on the perpendicular magnetization is out-of-plane and varies with the interfacial DMI constant, the applied in-plane magnetic fields, and the magnetic anisotropy distribution. Striking consequences of the effect include asymmetric current/field switching of perpendicular magnetization, hysteresis loop shift of perpendicular magnetization in the absence of in-plane direct current, and sharp in-plane magnetic field switching of perpendicular magnetization. Utilizing the intralayer DMI, we demonstrate programable, complete Boolean logic operations within a single spin-orbit torque device. These results will stimulate investigation of the long-range intralayer DMI effect in a variety of spintronic devices.

Electrical switching of magnetization is central to spintronic memory and computing[1-3]. Despite the enormous investigations in the past two decades[1-8], the understanding of in-plane current-induced switching of perpendicular magnetization remains elusive as indicated by a number of remarkable long-standing puzzles (see ref. 9 for a review of this problem). In many spin-orbit torque (SOT) heterostructures, such as heavy metal/ferromagnet (HM/FM) bilayers with perpendicular magnetic anisotropy (PMA), the scaling of the switching current density with the SOT and the applied magnetic field is in strong disagreement with the predictions of the existing macrospin and chiral-domain-wall depinning models[10,11]. For instance, strong asymmetry occurs in the switching current densities of many PMA heterostructures under the same in-plane assisting field[12-16]. The in-plane magnetic field hinders rather than assists current-driven switching of perpendicular

magnetization in some cases[17]. Quantitatively, analyses of the switching current density following the macrospin and chiral-domain-wall depinning models[10,11] typically overestimate or underestimate the SOT efficiency of a PMA heterostructure by up to thousands of times[9,18,19]. These remarkable puzzles suggest that some fundamental physics for electrical switching of magnetization is yet to be discovered.

Here, we report the discovery, the characteristics, and the applications of the long-range intralayer Dzyaloshinskii-Moriya interaction (DMI) effect, which is a new type of DMI that is distinct from the yet-known DMI effects (e.g., short-range interfacial DMI coupling of neighboring atomic spins within the same magnetic layer interfaced with a HM layer (Fig. 1a)[20-28], long-range interlayer DMI coupling of two orthogonal magnetic layers separated by a HM layer in in-plane FM/ HM/perpendicular FM trilayers (Fig. 1b)[29-35], or bulk DMI in thick FM

[1]State Key Laboratory for Superlattices and Microstructures, Institute of Semiconductors, Chinese Academy of Sciences, Beijing 100083, China. [2]College of Materials Science and Opto-Electronic Technology, University of Chinese Academy of Sciences, Beijing 100049, China. [3]Institute of Microelectronics, Chinese Academy of Sciences, Beijing 100029, China. [4]School of Integrated Circuits, University of Chinese Academy of Sciences, Beijing 100049, China. [5]Present address: School of Integrated Circuits, University of Chinese Academy of Sciences, Beijing 100049, China. ✉ e-mail: ljzhu@semi.ac.cn

**Fig. 1 | Schematics of the three types of DMI effects. a** Short-range interfacial DMI of heavy metal/ferromagnet (HM/FM) interfaces, describing the HM-mediated chiral coupling of neighboring atomic spins ($s_1$, $s_2$) within the same magnetic layer. **b** Long-range interlayer DMI in in-plane FM/HM/perpendicular FM trilayers, describing the HM-mediated chiral coupling of two orthogonal magnetic domains ($M_1$, $M_2$) separated by a heavy metal layer. **c** Long-range intralayer DMI in an anisotropy-fluctuated magnetic layer adjacent to a heavy metal, describing the HM-mediated chiral coupling of two orthogonal magnetic domains ($M_1$, $M_2$) separated by a magnetic domain wall. The red and gray arrows represent the local magnetic moments in the FM layers.

layers[36,37]). The long-range intralayer DMI describes the HM-mediated chiral coupling of two orthogonal magnetic domains separated by a magnetic domain wall within the same magnetic layer. The long-range intralayer DMI in PMA HM/FM heterostructures manifests as an effective perpendicular magnetic field ($H_{DMI}^z$, Fig. 1c) that promotes or hinders the switching of perpendicular magnetization, ultimately leading to a number of striking consequences, e.g., strong asymmetry in the switching density, hysteresis loop shift in the absence of in-plane direct current, and switching of perpendicular magnetization purely by an in-plane magnetic field. None of these characteristics can be attributed to the short-range interfacial DMI[6,9,11]. We also demonstrate that the long-range intralayer DMI effect provides a new platform for designing functional spintronic devices.

## Results

### Current-driven magnetization switching

Magnetic heterostructures for switching measurements (Fig. 2a) include Ir 5.4/FeCoB 1, W 4/FeCoB 1, Ta 5/FeCoB 1, Cr 5/Ti 1/FeCoB 1, Pt 5/Co 1, Ir 5/Co 1, and Pd 5/Co 1 with strong PMA (the numbers are layer thicknesses in nanometer, FeCoB = $Fe_{60}Co_{20}B_{20}$). As described in detail in the Sec. Method, these samples are sputter-deposited with our optimized growth protocol that typically yields reasonably sharp interfaces and no obvious intermixing or magnetic dead layer[36,38–40] (e.g., for Pt/Co samples see the results of scanning transmission electron microscopy and the thickness-dependent magnetic moment measurements in the Supplementary Fig. 1). These samples exhibit high magnetization and strong interfacial PMA energy density ($M_s \approx 1200$ emu/cm$^3$ for FeCoB and $\approx 1400$ emu/cm$^3$ for Co, $K_s \approx 1.1$–2.0 erg/cm$^2$, see Supplementary Table 1), which reaffirms the sharp interfaces of these samples since interfacial intermixing, if significant, would substantially degrade the apparent magnetization and the interfacial PMA[39,40].

We first show in Fig. 2b the results of current-induced magnetization switching in the Ir 5.4/FeCoB 1, the W 4/FeCoB 1, and the Pt 5/Co 1 bilayers under different $H_x$. The current densities for downwards and upwards switching are strongly asymmetric in magnitude for the same in-plane magnetic fields $H_x$. This is striking because both the macrospin and the domain-wall depinning models[9–11] suppose the current density for upward ($j_\uparrow$) and downward ($j_\downarrow$) magnetization switching to be of the same magnitude as soon as the applied in-plane magnetic field is also of the same magnitude (i.e., $|j_\uparrow| = |j_\downarrow|$ for a given $|H_x|$, see Fig. 2c for domain-wall depinning model). When $j_\uparrow$ and $j_\downarrow$ of the samples are plotted as a function of $H_x$ in Fig. 2d, it becomes evident that the switching current asymmetry can be described by a current shift, $j_{shift} = (j_\downarrow + j_\uparrow)/2$, that is independent of the current direction. The switching current densities after subtraction of the shift, $j_{\uparrow 0} = j_\uparrow - j_{shift}$ and $j_{\downarrow 0} = j_\downarrow - j_{shift}$, decrease as $|H_x|$ increases and remain symmetric for $\pm H_x$, in agreement with the expectation of anti-damping torque-driven depinning of chiral domain walls with the short-range interfacial DMI.

In the domain-wall depinning mechanism that typically dominates the current-induced switching process of PMA Hall bars (Fig. 2e), the nonzero $j_{shift}$ is equivalent to an effective perpendicular field that adds to or subtracts from the dampinglike torque field ($H_{DL}$) (which is $H_{DMI}^z$ as we verify below), i.e., $H_{DMI}^z \propto j_{shift}$ sign($\theta_{SH} H_x$). As plotted in Fig. 2f, $j_{shift}$sign($\theta_{SH} H_x$) varies significantly with $H_x$ and reverses sign when the orientation of $H_x$ is reversed. This effective perpendicular field is independent of the sign and the magnitude of $\theta_{SH}$ of the HM because the W and Ir samples with opposite spin Hall ratio signs ($\theta_{SH} < 0$ for W and $\theta_{SH} > 0$ for Ir[41]) have the similar $j_{shift}$ sign($\theta_{SH} H_x$).

### Magnetic field-driven magnetization switching

We also observe a significant asymmetry and shift effect in the magnetic field-driven magnetization switching experiments, in which there is no applied direct current and thus no spin-orbit torque. During the field switching experiment, a small sinusoidal electric field of $\approx 1.5$ kV/m is used as the excitation for the detection of the anomalous Hall signal using a lock-in amplifier. The anomalous Hall resistance hysteresis loops are measured by sweeping the external magnetic field in the $xz$ plane ($H_{xz}$) at different fixed polar angles ($\theta_H$) (see Fig. 3a for the data of the Ir/FeCoB sample). As summarized in Fig. 3c, the switching field ($H_{sw}^{xz}$) deviates from the $1/\cos\theta_H$ scaling asymmetrically around $\theta_H = \pm 90°$, which is contrary to the conventional chiral domain wall depinning model that assumes a critical role of the interfacial DMI, $H_{sw}^{xz} = H_{sw}^z / \cos\theta_H$, and a constant perpendicular coercivity ($H_{sw}^z$)[7,9,42]. It is also striking that the perpendicular magnetization exhibits a sharp full switching at $\theta_H = \pm 90°$ under an in-plane magnetic field (Fig. 3a).

The anomalous Hall hysteresis loops are then measured by sweeping the perpendicular magnetic field ($H_z$) under given in-plane magnetic fields ($H_x$). Strikingly, all the samples in this work exhibit a strong hysteresis loop shift in the absence of any direct current (see Fig. 3b for the results of a Ir 5.4/FeCoB 1 sample). In other words, the upward and downward perpendicular switching fields ($H_\uparrow^z$ and $H_\downarrow^z$) are different in magnitude under a given in-plane bias field $H_x$. As shown in Fig. 3d, the relative asymmetry in the switching fields is very strong for the Ir 5.4/FeCoB 1 and the W 4/FeCoB 1 but weak for the Pt 5/Co 1 (the absolute asymmetry is also strong, see below), which is consistent with the trend of the switching current asymmetry in current-driven switching (Fig. 2f). As shown in Fig. 3d, the perpendicular switching fields $H_\uparrow^z$ and $H_\downarrow^z$ also increase or decrease in magnitude as the in-plane magnetic field increases, suggesting the breakdown of the widely accepted assumption that the perpendicular coercivity $H_{sw}^z$ does not vary with the in-plane magnetic field (such that $H_{sw}^{xz}$ is supposed to follow the $1/\cos\theta_H$ scaling). As shown in Fig. 3e–g, a transverse magnetic field ($H_y$) also induces full magnetization switching and the same asymmetry as $H_x$, reaffirming the irrelevance to the current flow within the Hall bar. We also note that this is a distinct effect from the well-known direct current-induced loop shift[43] because the latter occurs due to the direct current-induced anti-damping SOT field exerted on the magnetic domain wall moment.

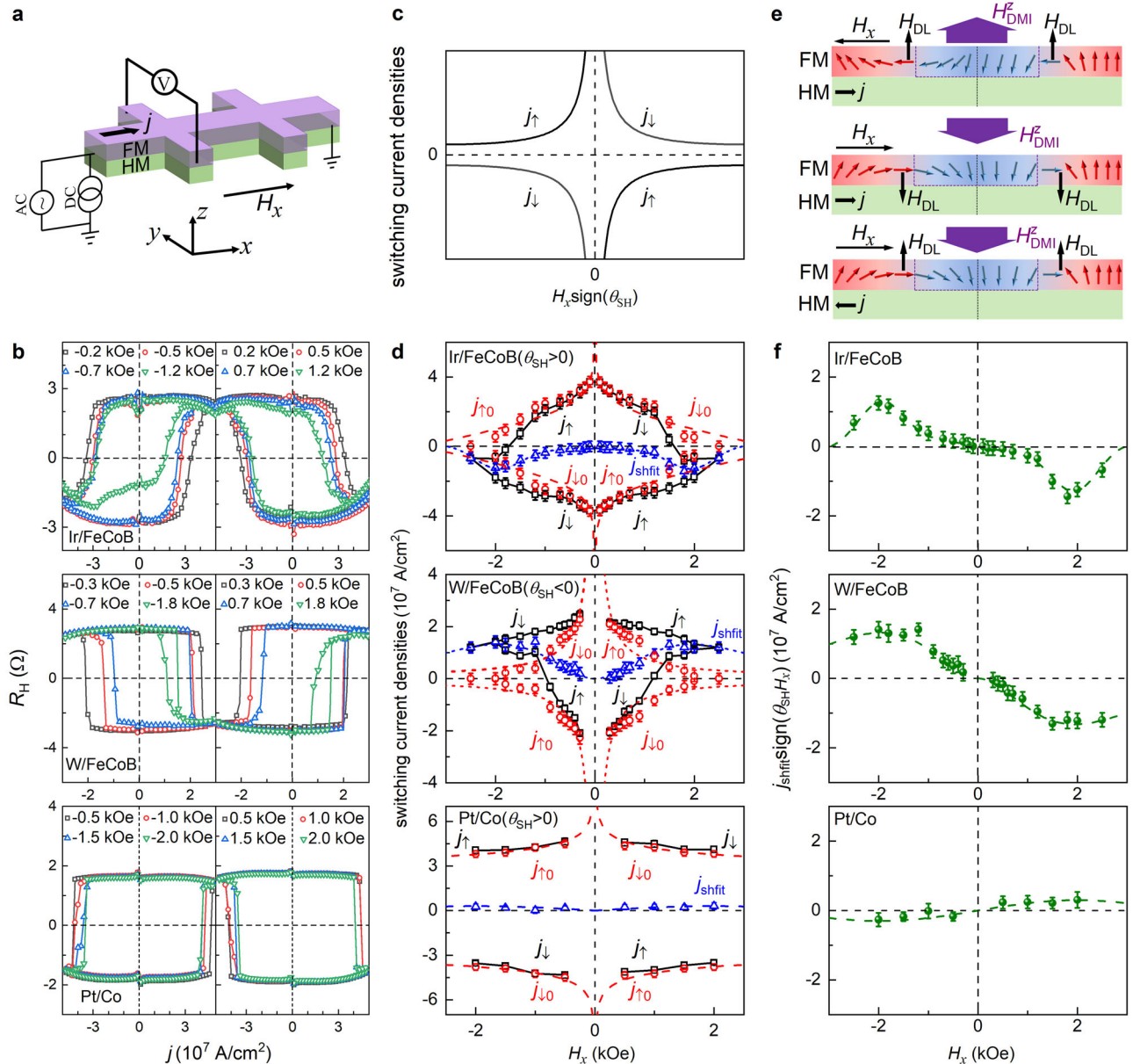

**Fig. 2 | Electrical switching behaviors. a** Experimental configuration of the current-driven magnetization switching. **b** Hall resistance vs the dc current density inside the HM layer for Ir 5.4/FeCoB 1, W 4/FeCoB 1, and Pt 5/Co 1 under positive and negative in-plane magnetic fields $H_x$. **c** Dependence on $H_x\mathrm{sign}(\theta_{SH})$ of the switching current density ($j_c$) predicted by the domain wall depinning model that considers $H_{DL}$ and the short-range interfacial DMI but assumed zero $H_{DMI}^z$. In this case, the upwards and downwards switching currents ($j_\uparrow$ and $j_\downarrow$) are of the same magnitudes under the in-plane longitudinal fields of the same magnitudes $|H_x|$. **d** The switching

current densities, $j_\uparrow, j_\downarrow, j_{shift} = (j_\downarrow + j_\uparrow)/2$, $j_{\uparrow 0} = j_\uparrow \cdot j_{shift}$, and $j_{\downarrow 0} = j_\downarrow \cdot j_{shift}$ for Ir 5.4/FeCoB 1, W 4/FeCoB 1, and Pt 5/Co 1 under different $H_x$. $j_{\uparrow(\downarrow)}, j_{\uparrow(\downarrow)0}$, and $j_{shift}$ are plotted using black squares, red circles, and blue triangles, respectively. **e** Schematic illustration of current-induced damping-like spin-orbit torque field ($H_{DL}$) and the long-range intralayer DMI field ($H_{DMI}^z$) on the adjacent perpendicular and in-plane magnetic moments under different orientations of $H_x$ and charge current. **f** $j_{shift}\mathrm{sign}(\theta_{SH}H_x)$ for Ir 5.4/FeCoB 1, W 4/FeCoB 1, and Pt 5/Co 1 under different $H_x$. In (**d**, **f**), the dash lines are to guide the eyes. Error bars represent standard deviations.

## Long-range intralayer DMI effect

Next, we show that the nature of the perpendicular magnetic field suggested by the asymmetric switching and the loop shift is most likely the effective field of a long-range intralayer DMI torque on the perpendicular magnetization of magnetic layers with finite magnetic anisotropy fluctuations. As schematically shown in Fig. 4a, a realistic magnetic layer typically has non-uniformity in its magnetic anisotropy, evidence of which includes the widely existing two-magnon scattering damping of in-plane magnetized HM/FM bilayers[44,45] and gradual, memristor-like, or even partial electrical switching behaviors of PMA samples[46,47] (see more examples in Fig. 2b, ref. 9 and references therein). Note that a perfectly uniform PMA sample should only have a

sharp two-state switching (only the $\pm M_z$ states). While the PMA samples with sizable anisotropy can remain perfectly magnetized as a macro-spin along the $+z$ or $-z$ directions at remanence states, the magnetic domains with lower anisotropy will be first tilted and aligned collinear with the magnetic field when the applied in-plane magnetic field increases from zero to a certain value, leading to formation of coexistence of perpendicular and in-plane domains and thus a lateral perpendicular FM/HM/in-plane FM configuration. For the PMA samples in this work, the embedment of the in-plane domain within the perpendicular FM host is readily seen from the polar magneto-optical Kerr effect (MOKE) microscopy images (see Fig. 4b–f for the W/FeCoB device and Supplementary Fig. 3 for the Ir/FeCoB sample). The white

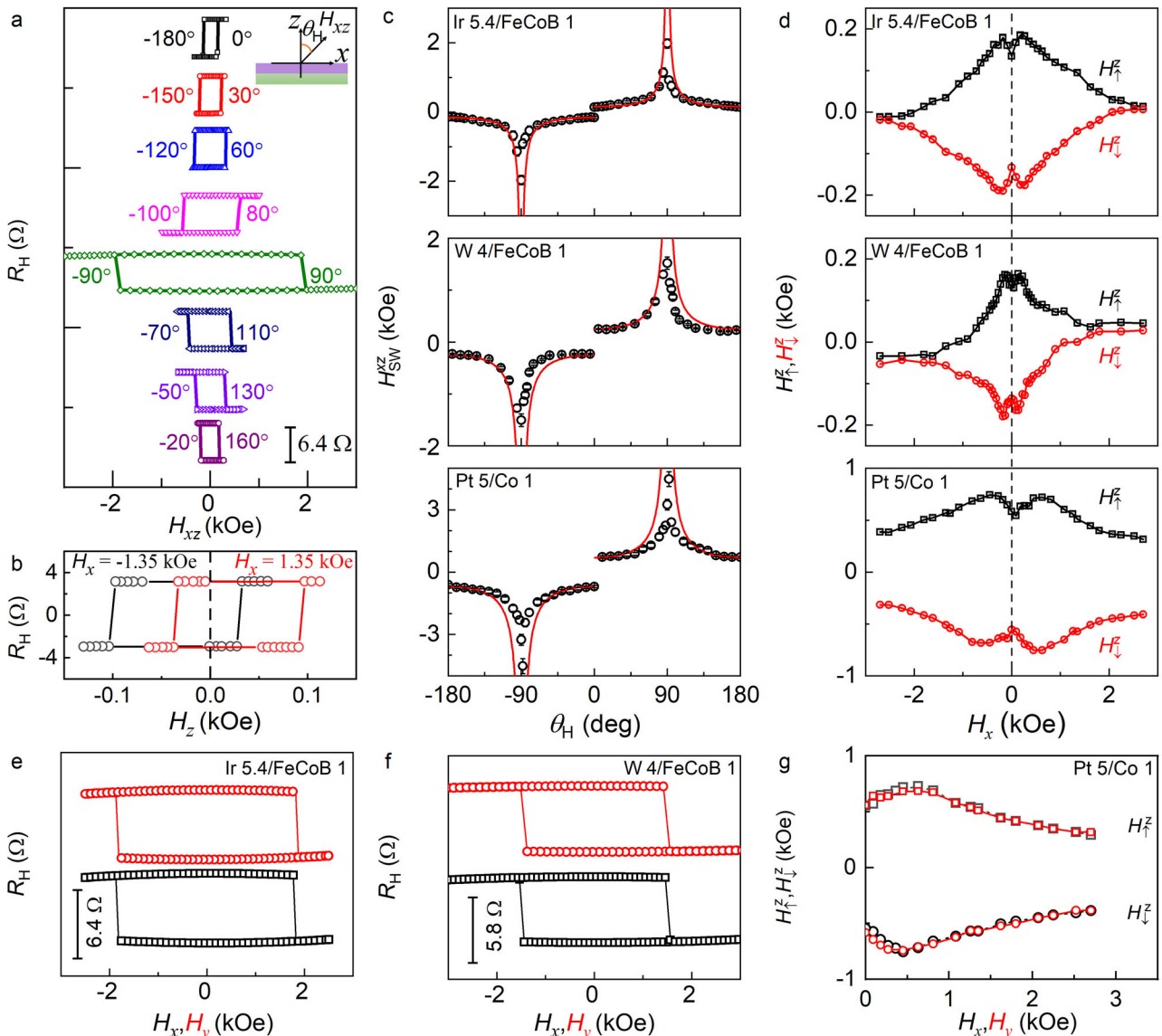

**Fig. 3 | Magnetic field switching behaviors. a** Hall resistance hysteresis loops of the Ir 5.4/FeCoB 1 driven by an external magnetic field in $xz$ plane ($H_{xz}$) at different polar angles ($\theta_H$), from which the switching field $H_{sw}^{xz}$ are determined. **b** Hall resistance hysteresis loops of the Ir 5.4/FeCoB 1 driven by the perpendicular magnetic field ($H_z$) under a fixed in-plane magnetic field of $H_x = 1.35$ kOe (black) and $-1.35$ kOe (red). **c** Dependence of $H_{sw}^{xz}$ on $\theta_H$ for the Ir 5.4/FeCoB 1, W 4/FeCoB 1, and Pt 5/Co 1. **d** Dependence of $H_\uparrow^z$ (black) and $H_\downarrow^z$ (red) on the in-plane magnetic field ($H_x$) for the Ir 5.4/FeCoB 1, W 4/FeCoB 1, and Pt 5/Co 1. Hall resistance hysteresis loops of (**e**) the Ir 5.4/FeCoB 1 and (**f**) the W 4/FeCoB 1 driven by an in-plane longitudinal magnetic field ($H_x$, black) and transverse magnetic field ($H_y$, red). **g** Good consistency of the upward and downward perpendicular switching fields ($H_\uparrow^z$ and $H_\downarrow^z$) for the Pt 5/Co 1 under longitudinal (black) and transverse (red) magnetic fields. In (**e**–**g**), the black squares are the data from the $H_x$-swept measurements, the red circles from the $H_y$-swept measurements. In (**e**) and (**f**), the hysteresis loops are shifted vertically for clarity. Error bars are standard deviations.

and black regions of the polar MOKE images are the perpendicular magnetic domains pointing to +$z$ and -$z$ directions (i.e., the +$M_z$ and -$M_z$ domains), respectively, while the gray Hall bar regions in Fig. 4c, e, which have the same contrast as the fully in-plane magnetized Hall bar by an in-plane magnetic field of +5.5 kOe in Fig. 4f and the substrate area with the magnetic stack etched away, are the in-plane domains due to the application of an in-plane field of $H_x = -1.66$ kOe (+1.66 kOe) to the perfect +$M_z$ state in Fig. 4b (-$M_z$ state in Fig. 4d). These observations provide direct evidence for the magnetic anisotropy non-uniformity in the magnetic heterostructures, i.e., some regions have weaker PMA than others and can be aligned in-plane by a smaller magnetic field.

The in-plane magnetic domain is expected to exert an effective perpendicular DMI field on the perpendicular domain,

in analog to the interlayer DMI of a vertical trilayer of perpendicular FM/HM/in-plane FM[30–33]. Specifically, the perpendicular moment $\mathbf{M}_2$ will be coupled to the adjacent in-plane moment $\mathbf{M}_1$ via a HM atom according to the Levy-Fert three-point model[21,23,48] (Fig. 4a), with the DMI energy of

$$E_{DMI} = \mathbf{D}_{12} \cdot (\mathbf{M}_1 \times \mathbf{M}_2), \quad (1)$$

or

$$E_{DMI} = -\mathbf{M}_2 \cdot (-\mathbf{D}_{12} \times \mathbf{M}_1), \quad (2)$$

where the DMI vector $\mathbf{D}_{12} = \zeta D\, \mathbf{r}_1 \times \mathbf{r}_2$. Here, $\mathbf{r}_{1,2}$ is the unit vector linking the mediate HM atom and interacting moment $\mathbf{M}_{1,2}$, $D$ is the interfacial

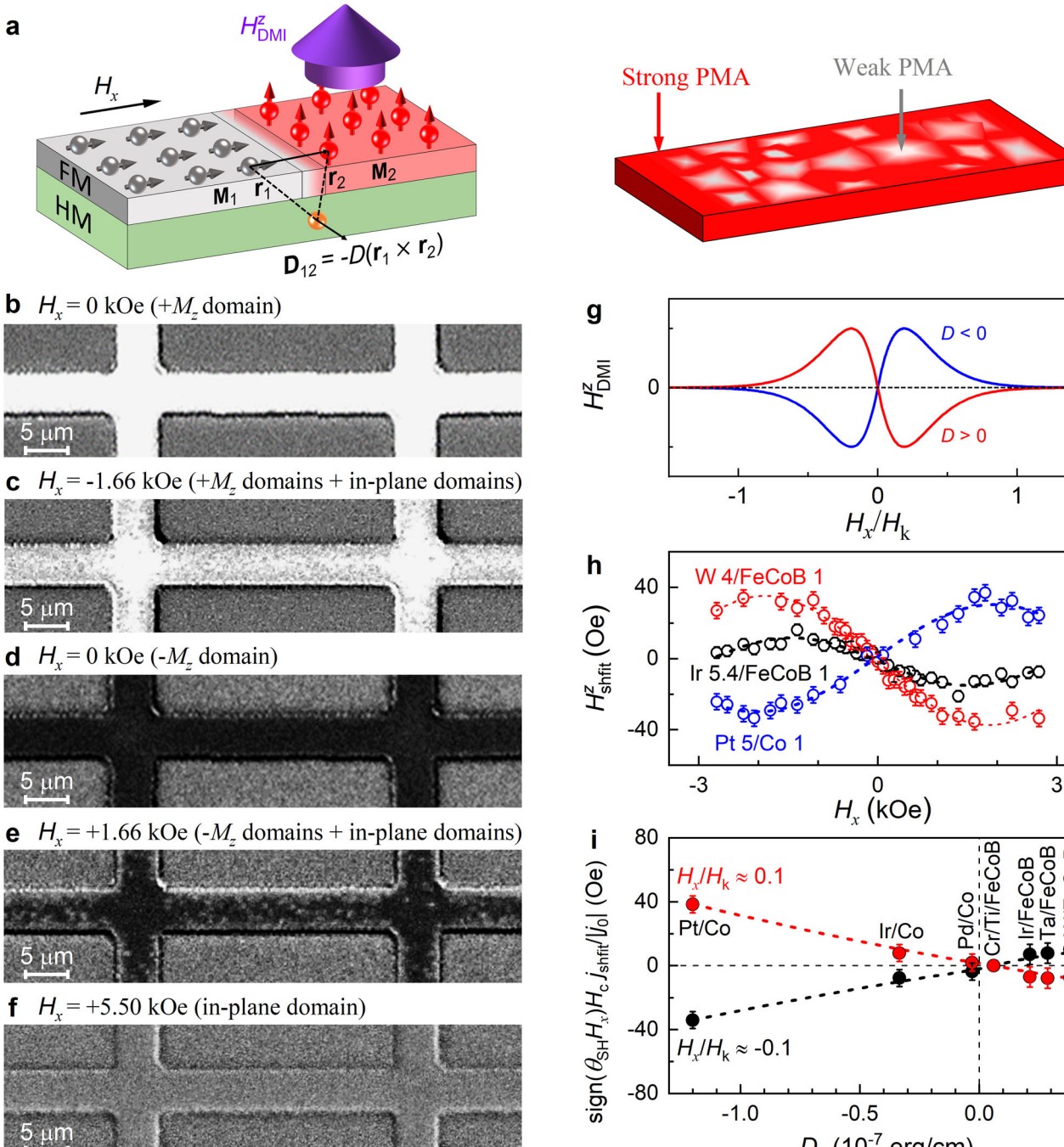

**Fig. 4 | Long-range intralayer DMI effect. a** Schematic illustration of the long-range intralayer DMI coupling between neighboring in-plane and perpendicular magnetic moments $M_1$ and $M_2$ via a heavy metal (HM) in a HM/FM bilayer due to the interplay of the magnetic anisotropy non-uniformity and the in-plane magnetic field. Polar MOKE images for a W/FeCoB Hall-bar device (**b**) with only the $+M_z$ state magnetic domain (white) under $H_x = 0$ kOe (the remanence state after saturation along +z direction), (**c**) with in-plane magnetic domains (gray, in the same color as the substrate area with magnetic stack etched away) embedded in the $+M_z$ state magnetic domains (white) after application of $H_x = -1.66$ kOe to the perfect $+M_z$ state in (**b**), **d** with only the $-M_z$ state magnetic domain (black) under $H_x = 0$ kOe (the remanence state after saturation along -z direction), (**e**) with in-plane magnetic domains (gray) embedded in the $-M_z$ state magnetic domains (black) after application of $H_x = +1.66$ kOe to the perfect $-M_z$ state in (**d**), (**f**) with only the in-plane

magnetic domain under in-plane magnetic field of +5.5 kOe. The coexistence of the perpendicular and in-plane domains under an in-plane magnetic field in (**c**, **e**) reveals the presence of magnetic anisotropy non-uniformity in the sample. In (**b**–**f**), the background contrast is kept essentially the same during the MOKE imaging. **g** Expected scaling of the long-range intralayer DMI field ($H^z_{DMI}$) with $H_x/H_k$ for $D > 0$ (red) and $D < 0$ (blue). **h** Experimentally measured values of $H^z_{shift}$ plotted as a function of in-plane magnetic field $H_x$ for the Ir 5.4/FeCoB 1 (black), the W 4/FeCoB 1 (red) and the Pt 5/Co 1 (blue). **i** Dependence of $\text{sign}(\theta_{SH}H_x)H_cj_{shift}/|j_0|$ on the interfacial DMI strength ($D_s$) for the Pt/Co, Ir/Co, Pd/Co, Ta/FeCoB, Ir/FeCoB, and W/FeCoB under similar $H_x/H_k$ (the red dots for $H_x/H_k \approx 0.1$; the black dots for $H_x/H_k \approx -0.1$), suggesting a correlation of the long-range intralayer DMI field and the DMI constant of the HM/FM interface. Error bars are standard deviations due to measurement uncertainty.

DMI constant of the HM/FM interface, $\zeta$ is a parameter related to the dimensions of the interacting domains since the intralayer DMI is expected to be the strongest between the spins near the domain wall boundaries and decays between spins further away within the in-plane and out-of-plane domains. For simplicity, we only consider the DMI coupling between the perpendicular $\mathbf{M}_2$ and the magnetic component that is orthogonal to $\mathbf{M}_2$ (noted as in-plane $\mathbf{M}_1$) because any perpendicular component ($\mathbf{M}_{\text{perpendicular}}$) of a tilted domain or domain wall will have no DMI coupling with $\mathbf{M}_2$ for any DMI effects (i.e., $\mathbf{M}_{\text{perpendicular}} \times \mathbf{M}_2 = 0$). The collection of the in-plane component of a magnetic domain, which is the only part at work for the intralayer DMI effect in this study, is effectively an in-plane macrospin for the perpendicular domain $\mathbf{M}_2$. The effective magnetic field of the long-range intralayer DMI exerted on the perpendicular moment $\mathbf{M}_2$ via the in-plane moment $\mathbf{M}_1$ is $\mathbf{H}_{\text{DMI}} = -\mathbf{D}_{12} \times \mathbf{M}_1$, which has a perpendicular component of

$$H_{\text{DMI}}^z = -\zeta D M_1 \sin\beta, \tag{3}$$

where $\beta$ is the azimuth angle of $\mathbf{D}_{12}$ relative to the in-plane moment $\mathbf{M}_1$.

Equation (3) predicts $H_{\text{DMI}}^z$ to depend on the in-plane magnetic moment, interfacial DMI constant of the HM/FM interface, dimensions of the interacting domains, and the spatial location of $\mathbf{M}_1$ relative to $\mathbf{M}_2$. Thus, the reversal of the in-plane moment by an in-plane magnetic field should also change the sign of $H_{\text{DMI}}^z$ (Fig. 4g). As the external in-plane magnetic field increases in magnitude, $|H_{\text{DMI}}^z|$ should first increase gradually from essentially zero to a peak value when the in-plane magnetic field maximizes the effect via the growth of the in-plane magnetic domain $\mathbf{M}_1$, and then starts to decrease slowly towards zero as the in-plane field further increases towards the in-plane saturation field (the anisotropy field $H_k$) due to the reduction of the perpendicular domain ($\mathbf{M}_1 // \mathbf{M}_2$). While the effects of the dimensions and relative location of the interacting domains on the magnitude of $H_{\text{DMI}}^z$ still lacks a simple, unified analytical calculation due to the involvement of multiple complex microscopic configurations of the practical samples (see more discussions in the Supplementary Note 1), the nonzero total DMI effects in practical samples are proved by the consequences of various DMI effects (e.g., interfacial DMI-induced frequency difference between counterpropagating Damon-Eshbach spin waves[49,50], interlayer and intralayer DMI-induced switching asymmetry[30–35]).

The long-range intralayer DMI field would manifest as a shift in the critical switching current ($j_{\text{shift}}$) during a current switching experiment and a shift in the out-of-plane switching magnetic field, i.e., $H_{\text{shift}}^z = (H_\uparrow^z + H_\downarrow^z)/2$, in a magnetic field switching experiment. First, $j_{\text{shift}}\text{sign}(\theta_{\text{SH}}H_x)$ in Fig. 2f exhibits a sign change upon reversal of the in-plane field $H_x$ or reversal of the $D$ sign (W vs Pt) and varies with increasing magnitude of $H_x$, which are consistent with the characteristics of the long-range intralayer DMI. In Fig. 4h, we further show the values of $H_{\text{shift}}^z$ for the W 4/FeCoB 1, the Ir 5.4/FeCoB 1, and the Pt 5/Co 1 as a function of $H_x$ as determined from the field switching data in Fig. 3b,d. $H_{\text{shift}}^z$ exhibits all the characteristics of long-range intralayer DMI, including a sign change upon reversal of the in-plane field $H_x$ or reversal of the $D$ sign (W vs Pt), and the non-monotonic dependence on the in-plane magnetic field $H_x$.

To further test the correlation of the switching asymmetry to the DMI effect, we show in Fig. 4e the normalized switching current density asymmetry, $\text{sign}(\theta_{\text{SH}}H_x)H_c j_{\text{shift}}/|j_0|$, as a function of the interfacial DMI constant $D_s = D t_{\text{FM}}$ for various magnetic heterostructures ($t_{\text{FM}}$ is the thickness of the magnetic layer), including Ir 5.4/FeCoB 1, W 4/FeCoB 1, Ta 5/FeCoB 1, Cr 5/Ti 1/FeCoB 1, Pt 5/Co 1, Ir 5/Co 1 and Pd 5/Co 1 (see Supplementary Fig. 2 for the current switching data). Here, the $D_s$ values are adapted from previous reports on corresponding magnetic interfaces that have similar sharpness and interfacial PMA energy density as our samples do in this study (i.e., $D_s$ is $0.21 \times 10^{-7}$ erg/cm for Ir/FeCoB[49], $0.38 \times 10^{-7}$ erg/cm for W/FeCoB[51], $0.22 \times 10^{-7}$ erg/cm for Ta/FeCoB[52], $0.06 \times 10^{-7}$ erg/cm for Ti/FeCoB[53], $1.2 \times 10^{-7}$ erg/cm for Pt/Co[38], $0.34 \times 10^{-7}$ erg/cm for Ir/Co[49] and $0.01 \times 10^{-7}$ erg/cm for Pd/Co[54], also see Supplementary Table 1). Note that quantification of the $D_s$ values of these PMA samples from standard Brillouin light scattering (BLS) or loop shift measurements is prevented because the electromagnet of BLS setups available to us ($\leq 2$ kOe)[38,55] cannot overcome the strong PMA field of our samples (up to 10 kOe, see Supplementary Table 1) to align the magnetization in-plane for the BLS analysis and because the strong dependence of the switching field on the in-plane field at zero dc current (Fig. 3b) invalidates the loop shift technique for these samples. For all the heterostructures, the current density asymmetry increases with the magnitude of $D_s$, while it reverses the sign for positive $D_s$ compared to the negative $D_s$ case. Such a direct correlation of the switching asymmetry to the sign and strength of the interfacial DMI further reaffirms that the long-range intralayer DMI is the mechanism of the observed switching asymmetry. This conclusion is qualitatively robust and unaltered by any uncertainty of the used $D_s$ values.

We also note that the occurrence of the long-range intralayer DMI is interesting but not too surprising in terms of the chiral coupling distance. When the effective PMA field $H_k$ is much greater than any applied in-plane magnetic field, the domain wall width ($\Delta$) of an ultrathin PMA sample can be estimated by[56–60]

$$\Delta \approx \sqrt{A/(H_k M_s/2 + 2\pi N M_s^2)} \tag{4}$$

where $2\pi N M_s^2$ is the magnetostatic shape anisotropy of the domain wall and the demagnetizing factor $N$ is approximately $(1 + \Delta/t_{\text{FM}})^{-1}$ for the Bloch domain wall[56]. Theories and simulations have also indicated that $\Delta$ in ultrathin FMs is essentially independent of the DMI[61] and the domain wall configuration (Bloch or Néel)[62]. Therefore, Eq. (4) has been widely applied to various PMA heterostructures with DMI[56–60]. Note that the simplified relation $\Delta_{\text{upper}} \approx \sqrt{2A/H_k M_s}$ in the literature[63] has ignored the domain wall shape anisotropy and only yields the upper limit of the domain wall width. Using Eq. (4) and the exchange stiffness $A$ of $\approx 1.0$ μerg/cm for Co[64–66] and $\approx 0.8$ μerg/cm for FeCoB[67,68], we estimate the width of the magnetic domain wall that separates the domains coupled by the HM-mediated intralayer DMI layer as 3–6 nanometers for the samples in this study (Supplementary Table 1). These $\Delta$ values agree well with those estimated for typical PMA HM/FM heterostructures in the literature reports[56,57,60–63] and are within the typical range of the DMI effects[32,35,48,69,70]. For example, significant interlayer DMI coupling has been reported between neighboring orthogonal domains separated by a heavy metal layer that is typically several nm thick[32,35,69,70]. A long-distance chiral coupling is also suggested in an early experiment on the chiral coupling of two perpendicular domains via a deliberately fabricated in-plane magnetic domain of up to 200 nm long (which was reported before the discovery of the long-range DMIs and attributed to the short-range interfacial DMI)[48]. Note that direct experimental quantification of such few-nm magnetic domain width between the in-plane and perpendicular domains of strong PMA samples has been very challenging and beyond the scope of this work. To the best of our knowledge, there has been no report of a microscopic technique that simultaneously had a sensitivity capable of the very weak magnetic signal of the narrow domain walls of only 1 nm thick magnetic layer, a magnetically spatial resolution of ~1 nm or below, and an in-plane magnetic field of 1–3 kOe to form a magnetic domain wall between adjacent in-plane and perpendicular domains within strong PMA samples.

## Complete Boolean logic operations in a single device

Now we demonstrate that the five basic logic gates (AND, OR, NOT, NOR, and NAND) can be achieved in a single device utilizing the long-range intralayer DMI-induced asymmetry. As shown in Fig. 5a, the logic device consists of two parallel current inputs $I_A$ and $I_B$ and a Hall cross

of the W 4/FeCoB 1 bilayer. The logic value of the storing FeCoB layer is defined as "1" at the upward magnetization state ($+M_z$) and "0" at the downward magnetization state ($-M_z$). As shown in Fig. 5b, the switching of this Hall cross requires a total required current ($I_A + I_B$) of 2 mA (−0.9 mA) for upward switching and −0.9 mA (2 mA) for downward switching under $H_x = +0.6$ kOe (−0.6 kOe). For the AND and NAND logic gates (Fig. 5c), we define both $I_A$ and $I_B$ as "1" at 1.5 mA and as "0" at 0 mA and reset the gate state using a current pulse $I_A + I_B = −3$ mA. The device functions as an AND gate under the bias field $H_x = +0.6$ kOe and returns "1" only when the $I_A$ and $I_B$ inputs are "1" at the same time. In contrast, under the bias field $H_x = −0.6$ kOe, the device functions as a NAND gate and returns "0" only when the $I_A$ and $I_B$ inputs are both "1". The NAND gate is reduced to a NOT gate when $I_A$ is fixed at "1".

For the NOR and OR gates (Fig. 5d), $I_A$ and $I_B$ are defined as "1" at −1.5 mA and as "0" at 0 mA, while the gate is reset by a current pulse of 3 mA. The device is a NOR gate under $H_x = 0.6$ kOe and always returns "0" unless the $I_A$ and $I_B$ inputs are both "0". The device functions as a OR gate under $H_x = −0.6$ kOe and returns "1" except when $I_A$ and $I_B$ are both "0". We have also achieved all five Boolean logic operations utilizing other strong intralayer-DMI HM/FeCoB bilayers (e.g., Ta/FeCoB and Ir/FeCoB) with significant long-range intralayer DMI. Importantly, this represents the breakthrough achievement of the complete set of Boolean logic operations within a single SOT device using a constant-magnitude magnetic field. When an on-chip in-plane nanomagnet switchable by the SOT provides the constant-magnitude magnetic field, such SOT logic device utilizing the long-range intralayer DMI can be electrically operated without the need for any external magnetic field. The simple device architecture also allows for high-sensitivity readout using the tunnel magnetoresistance (TMR) of a magnetic tunnel junction. Therefore, from a technologic point of view, the long-range intralayer DMI-based multifunctional SOT logic device we propose here is highly preferred by large-scale integration into computing circuits and advantageous over the previously reported domain wall logic device driven by three orthogonal magnetic fields[71], the PMA Pt/Co[72] or Ta/CoFeB[73] Hall-bar devices driven by SOT and two varying-in-

magnitude magnetic fields, and the multi-state ferromagnetic multilayers[74] that can be readout by the anomalous Hall voltage (too small to control the opening/closing of a field effect transistor[75]) but hardly by the sensitive TMR.

## Discussion

We have presented the discovery of the long-range intralayer DMI effect in magnetic layers adjacent to heavy metals, i.e., the HM-mediated chiral coupling of two orthogonal magnetic domains separated by a magnetic domain wall within the same magnetic layer. The intralayer DMI exerts a strong out-of-plane effective magnetic field ($H_{DMI}^z$) on the perpendicular magnetization, in analog to the interlayer DMI of the FM/HM/FM trilayers[30–33]. $H_{DMI}^z$ varies with the sign/magnitude of the interfacial DMI constant, the applied in-plane magnetic field, and the uniformity of perpendicular magnetic anisotropy. Scientifically, $H_{DMI}^z$ leads to a variety of striking puzzles in magnetization switching, such as the strong asymmetry in current densities for SOT switching of magnetic heterostructures under the same in-plane field, an in-plane field dependent shift of anomalous Hall hysteresis loop in the absence of a direct current, and sharp switching of perpendicular magnetization by a pure in-plane magnetic field. The discovery of $H_{DMI}^z$ may also explain the puzzling breakdown of the macrospin approximation in low-field harmonic Hall voltage experiment on some PMA HM/FM bilayers, as indicated by the non-parabolic in-plane field dependence of the first harmonic Hall voltage and the non-linear in-plane magnetic field dependence of second harmonic Hall voltage[76,77]. This long-range intralayer DMI effect is also expected to exert an in-plane magnetic field on the in-plane moment via the perpendicular moments, providing a mechanism for the asymmetry in the switching current density of in-plane spin-orbit torque magnetic tunnel junctions[78]. Since the DMI can also be mediated by oxides (e.g., at FM/oxide interfaces), we speculate that such long-range intralayer DMI may also occur in FM/oxide bilayers without a HM. Compared to the interlayer DMI, the long-range intralayer DMI is more integration-friendly because it is based on the HM/FM bilayers that are most

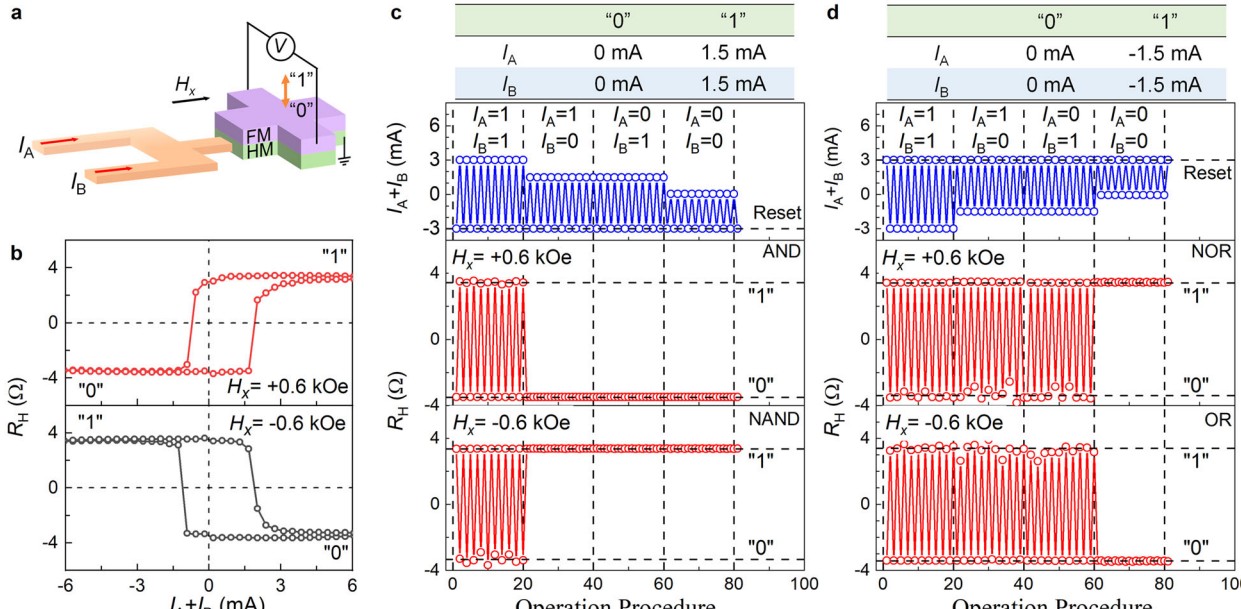

**Fig. 5 | Complete Boolean logic operations enabled by the long-range intralayer DMI in a single device. a** Schematic of a HM/FM logic device, with two current inputs ($I_A$ and $I_B$) and Hall resistance detection of "0" and "1" states. **b** Current-driven switching between the "0" and "1" states of a W 4/FeCoB 1 device under in-plane magnetic fields of $H_x = \pm 0.6$ kOe, suggesting a total switching current $I_A + I_B$ of −0.9 mA and 2 mA, respectively. **c** Programmable operation of AND

($H_x = +0.6$ kOe) and NAND ($H_x = −0.6$ kOe) gates. Inputs $I_A$ and $I_B$ are 0 mA for input "0" and 1.5 mA for input "1". **d** Programmable operation of NOR ($H_x = +0.6$ kOe) and OR ($H_x = −0.6$ kOe) gates. Inputs $I_A$ and $I_B$ are 0 mA for input "0" and −1.5 mA for input "1". Before each operation, the device is reset to the initial state by the reset current of $I_A = I_B = \pm 1.5$ mA. The dash lines are to guide the eyes.

compatible with SOT-MTJs[2,8]. We have demonstrated complete Boolean logic operations in a single device enabled by the long-range intralayer DMI effect. These findings will stimulate the investigation of long-range intralayer DMI and its impacts on a variety of magnetic heterostructures and devices.

In addition, we have also observed an unexpected increase and reduction of the perpendicular coercivity due to the in-plane magnetic field (Fig. 3d, g), which provides a possible underlying mechanism of the unusual increase of the switching current density $j_c$ with increasing in-plane magnetic field[17] as well as of the widely existing, remarkable under- or over-estimation of dampinglike torque efficiency[9,18,19] by the domain wall depinning analyses which assumed the perpendicular coercivity to be independent of in-plane magnetic field during current driven switching of PMA heterostructure[11].

## Methods
### Sample fabrications
Magnetic heterostructures of Ir 5.4/FeCoB 1, W 4/FeCoB 1, Ta 5/FeCoB 1, Cr 5/Ti 1/FeCoB 1, Pt 5/Co, Ir 5/Co 1, and Pd 5/Co 1 are sputter-deposited on oxidized Si substrates (the numbers are layer thicknesses in nanometer, FeCoB = $Fe_{60}Co_{20}B_{20}$). Each sample is seeded by a 1 nm Ta layer for improved adhesion and smoothness and protected from oxidization by a MgO 1.6/Ta 1.6 bilayer that is fully oxidized upon exposure to the atmosphere. The capping bilayer also enhances the perpendicular magnetic anisotropy of the magnetic layers. Each layer was sputter-deposited at a low rate (e.g., ≈ 0.007 nm/s for Co and FeCoB, ≈ 0.013 nm/s for Ir, ≈0.011 nm/s for W, ≈0.014 nm/s for Pt, ≈0.033 nm/s for Ta, ≈0.009 nm/s for Cr, ≈0.012 nm/s for Pd, ≈0.005 for Ti, and ≈0.004 nm/s for MgO). The base pressure during deposition is below $5 \times 10^{-9}$ Torr. These samples are patterned into $5 \times 60$ $\mu m^2$ Hall bars by photolithography and ion milling, followed by deposition of Ti 5/Pt 150 as the electrical contacts for switching measurements.

### Measurement
The saturation magnetization was measured by a superconducting quantum interference device. Direct and alternating currents are sourced into the Hall bars by a Keithley 6221 or by a SR860 lock-in amplifier, and the Hall voltage of the Hall bars is detected by a SR860 lock-in amplifier. The magnetic domains were imaged using magneto-optical Kerr effect microscopy at room temperature.

## Data availability
The data that support the findings of this study have been included in the maintext and the Supplementary Materials. Source data are provided with this paper.

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

## Acknowledgements

This work was supported partly by the National Key Research and Development Program of China (2022YFA1204004), by the Beijing Natural Science Foundation (Z230006), by the Strategic Priority Research Program of the Chinese Academy of Sciences (XDB44000000), and by the National Natural Science Foundation of China (12274405, 12304155, 62074164, and 62374180).

## Author contributions

L.Z. conceived the project, Q.L. fabricated the samples and performed the transport measurements, G.X. and L.L. performed the polar MOKE imaging, L.Z. and Q.L. wrote the manuscript, all the authors discussed the results and contributed to the manuscript writing.

## Competing interests

The authors declare no competing interests.
