## [Peer Review File · Nature Communications]

Asymmetric magnetization switching and programmable complete Boolean logic enabled by long-range intralayer Dzyaloshinskii-Moriya interactionEditorial Note: Parts of this Peer Review File have been redacted as indicated to remove third-party material where no permission to publish could be obtained.

Reviewers' comments:

Reviewer #1 (Remarks to the Author):

The authors have demonstrated current-induced SOT switching in FM/HM heterostructures in this work. They claim a novel DMI interaction named intralayer DMI is responsible for asymmetry in switching. They have shown some logic operation. There are several points the authors need to clarify before it can be considered for publication.

1. The claim that this is the first work to show the complete set of Boolean logic operations in a single device is not true. Several works demonstrate reconfigurable logic operations using a single device by magnetic field and SOT switching. The authors need to do a thorough literature survey before making such claims. Please see these references <https://www.nature.com/articles/srep20130> and <https://onlinelibrary.wiley.com/doi/abs/10.1002/aelm.201901090> of course, you will find a few more in the literature.
2. The Kerr microscopy images to claim the presence of in-plane and out-of-plane domain at 1.66K Oe is unclear. I am unable to see three different contrasts as claimed in the manuscript. It could be a state a multi-domain switching that is yet to be completed. The authors should give a scale bar. Why not present a high-resolution image with little more zoom in to substantiate the claims
3. Is there any role of capping layer MgO (1.6 nm) in these observations? Why MgO? why not just Ta 1.6 nm which would anyway completely oxidize.
4. The authors have shown only one set of inputs (1 and 1) to demonstrate NAND and AND gate operation. However, they should demonstrate all four possible configurations (0 and 0) which need to result in 1 as output for demonstrating NAND gate operation and 0 for AND gate operation.
5. The authors need to proofread the manuscript thoroughly. There are a few typos and grammatical mistakes in the text.

Reviewer #2 (Remarks to the Author):

The authors present a novel type of DMI – intralayer one – as a missing element among the DMI family. However, intralayer DMI was introduced in 2019 in [Nat. Mater. 18, 679–684 (2019). <https://doi.org/10.1038/s41563-019-0386-4>] as an interaction, which favors one sense of rotation of spins in the same FM layer. In addition, a novel RKKY-type DMI has been found in synthetic ferromagnets [Nat Commun 12, 3280 (2021), <https://doi.org/10.1038/s41467-021-23586-y>; Nano Letters 2023 23 (18), 8690-8696, <https://doi.org/10.1021/acs.nanolett.3c02607>]. Recently, the DMI family has been widened by compositional gradient-induced DMI [Acta Materialia 241, (2023) 118383, <https://doi.org/10.1016/j.actamat.2022.118383>; Nano Lett. 2022, 22, 24, 10128–10133, <https://doi.org/10.1021/acs.nanolett.2c03973>].

The idea presented in the manuscript looks interesting, but the experimental evidence and

explanation are still puzzling. The three-side model used describes the interfacial DMI, where the interacting spins can have any direction. In the present paper the interacting spins are orthogonal because they are in the neighboring domains with different magnetization vectors; one is in-plane and another is out-of-plane. However, in a real system two domains are separated by a finite-width domain wall, but the model does not take it into account. In my opinion, the discovered effect originates from an interfacial DMI, which brings magnetization switching asymmetry. To prove the intralayer origin, additional experiments are needed to be done. First, BLS measurements are crucial to probe spin wave dynamics and extract the DMI value and sign for each layered system under investigation. It is not correct to operate DMI values from literature because it is well-known that DMI is extremely sensitive to the crystal structure and quality of interfaces. Second, the precise structural properties including a cross-sectional view with subatomic resolution is required. It helps to estimate interfaces and the atomic structure. Third, magnetic properties have to be measured and collate with the structure including magnetically dead layers. It gives the effective M_s and anisotropy. The effect of DMI on magnetization switching cannot be discuss without whole understanding of all energy contributions.

In conclusion, I'd like to recommend conducting a comprehensive investigation in order to prove the existence of the proposed intralayer DMI, because now it seems just like a conventional interfacial DMI.

Minor remarks

1. Kerr microscope images with complete in-plane saturation magnetization are needed as an addition to Fig.4b.
2. A detailed description of the sample fabrication process is required.

Reviewer #3 (Remarks to the Author):

The authors report the discovery of the "intralayer Dzyaloshinskii-Moriya interaction (DMI) effect". They argued that the intralayer DMI is distinct from the already-known DMI inside a magnetic layer (also intralayer DMI). The "new" intralayer DMI refers to the DMI interaction among adjacent magnetic domains with different anisotropies that favor either in-plane or out-of-plane alignment. The manuscript presents current-induced switching experiments of perpendicular magnetization in different kinds of samples with interfacial DMI and an inhomogeneous perpendicular magnetic anisotropy. Due to the following reasons, the manuscript is unsuitable for Nature Communications.

The polar MOKE measurements reveal that there is a magnetic anisotropy non-uniformity in one of the samples (W/CoFeB), with some domains with low perpendicular anisotropy and others with large perpendicular anisotropy. This is a central issue in the manuscript since the proposed intralayer DMI relies on the anisotropy non-uniformity. However, there needs to be more information about the sizes of the domains and the domain walls that separate them. This is important since the interfacial DMI is a type of exchange interaction among neighboring magnetic moments, with a spatial extension of some nanometers, which is relatively small compared to the domain wall size. Then, I am trying to understand why the authors assume an interfacial DMI considering each magnetic domain as a single spin (see Eq. 3), which means the interfacial DMI is active on a hundred nanometers scale. This is an extreme and unjustified assumption that makes the interpretation of the results very hard to believe.

Response letter

We thank all the three referees for the comments and suggestions, following which we have majorly revised our manuscript. We believe these revisions have adequately addressed all the comments of the referees and have made our manuscript more compelling. We have provided manuscript with all the revisions yellow-highlighted. Below we provide the point-by-point responses to the comments of the three referees in the response letter (page 1-3 for the responses to Referee #1, pages 4-7 to Referee #2, and pages 8-10 to Referee #3).

Reviewer #1 (Remarks to the Author):

Comment 1-1: The authors have demonstrated current-induced SOT switching in FM/HM heterostructures in this work. They claim a novel DMI interaction named intralayer DMI is responsible for asymmetry in switching. They have shown some logic operation. There are several points the authors need to clarify before it can be considered for publication.

1. The claim that this is the first work to show the complete set of Boolean logic operations in a single device is not true. Several works demonstrate reconfigurable logic operations using a single device by magnetic field and SOT switching. The authors need to do a thorough literature survey before making such claims. Please see these references <https://www.nature.com/articles/srep20130> and <https://onlinelibrary.wiley.com/doi/abs/10.1002/aelm.201901090>. of course, you will find a few more in the literature.

Response: Following the suggestion of the referee, we have performed a more thorough literature investigation on logic operations in a single device driven by magnetic field and SOT and cited several more references that are relevant (refs. 63-66). However, none of the logic devices in the literature can be programmed to function the complete set of the five basic Boolean logic operations within a single device using a constant-magnitude magnetic field. For example, the Scientific reports paper (<https://www.nature.com/articles/srep20130>) reported the complete set of logic operations in a domain wall logic device driven by three different magnetic fields; the Advanced Electronic Materials paper (<https://onlinelibrary.wiley.com/doi/abs/10.1002/aelm.201901090>) included no demonstration of NOT and NAND. Therefore, our comment that “**this represents the first achievement of the complete set of Boolean logic operations within single SOT device using a constant-magnitude bias magnetic field**” holds. This emphasize that enabling of complete set of Boolean logic operations with a constant bias field magnitude is technologically advantageous, because tuning of the field magnitude in a circuit is challenging but a constant magnetic field can be provided simply by the stray field of an in-plane magnetization switchable by a spin torque.

In our revised manuscript, we have improved our discussions: “**We note that this represents the first achievement of the complete set of Boolean logic operations within a single SOT device using a constant-magnitude bias magnetic field. When an on-chip in-plane nanomagnet switchable by the SOT provides the constant-magnitude bias magnetic field, such SOT logic device utilizing the long-range intralayer DMI can be electrically operated without the need of any external magnetic field. The simple device architecture also allows for high-sensitivity readout using the tunnel magnetoresistance (TMR) of a magnetic tunnel junction. Therefore, from a technologic point of view, the long-range intralayer DMI-based multifunctional SOT logic device we propose here is highly preferred by large-scale integration into computing circuits and advantageous over the previously reported domain wall logic device driven by three orthogonal magnetic fields,⁶³ the PMA Pt/Co⁶⁴ or**

Ta/CoFeB⁶⁵ Hall-bar devices driven by SOT and two varying-in-magnitude magnetic fields, and the multi-state ferromagnetic multilayers⁶⁶ that can be readout by the anomalous Hall voltage (too small to control the opening/closing of a field effect transistor⁶⁷) but hardly by the sensitive TMR.”

Comment 1-2: 2. The Kerr microscopy images to claim the presence of in-plane and out-of-plane domain at 1.66 KOe is unclear. I am unable to see three different contrasts as claimed in the manuscript. It could be a state a multi-domain switching that is yet to be completed. The authors should give a scale bar. Why not present a high-resolution image with little more zoom in to substantiate the claims.

Response: Following the suggestion of the referee, we have provided a little more zoom-in Kerr microscopy images with a scale bar and notes of domain types in Fig.4b-f and Extended Data Fig. 3a-c of our revised manuscript. There are clearly three contrasts if the images of the W/FeCoB sample in Fig.4b-f are considered together (the background contrast was kept essentially the same during the MOKE imaging): the white and black in Kerr microscopy images represent the M pointing to the $+z$ and $-z$ axes, while the gray domains in Fig. 4c,e,f are in-plane domains and have the same contrast as the substrate with the magnetic layer etched away. In our revised manuscript, we have extended the description and the discussions of Kerr microscopy images in the figure caption and in the main text. From large-area Kerr microscopy images for the Ir 5.4/FeCoB 1 in Extended Data Fig. 3, one can see more clearly the coexistence of $+M_z$ domains (white), $-M_z$ domain (black), and in-plane domains (gray) during switching.

These Kerr microscopy images should be sufficiently clear for this purpose and are already the best our Kerr microscopy (imported from Singapore by Guozhong Xing) can obtain. If the referee still found the image blurred, it might be simply because the image quality is significantly lowered when the manuscript was converted into the small size pdf file for the review process.

In our view, the embedded gray regions in the white or black host domains in the MOKE images under $H_x = \pm 1.66$ kOe are not likely to represent a multidomain switching of a simple PMA sample, this is because in that case the domains should be pointing to the $+z$ or the $-z$ axes (black or white if the background contrast is kept essentially the same during the MOKE imaging as we do). We also emphasize that, without the presence of the low-anisotropy domains and a significant intralayer DMI, one cannot provide a mechanism for the striking magnetization switching effects, including the strong asymmetry in the switching density, hysteresis loop shift in the absence of in-plane direct current, and switching of perpendicular magnetization purely by an in-plane magnetic field.

3. Is there any role of capping layer MgO (1.6 nm) in these observations? Why MgO? why not just Ta 1.6 nm which would anyway completely oxidize.

Response: The capping MgO/Ta bilayers protects the magnetic layers from oxidation and enhances the perpendicular magnetic anisotropy, which are useful for the study of the spin-torque switching of the perpendicular magnetization, understanding of which is the key task of our present manuscript. A 1.6 nm would be too thin to protect the magnetic layers Co and FeCoB, while a too thick Ta might not be fully oxidized and cause shunting of current and add complications to the analysis of the switching experiment.

In Method section of our revised manuscript, we have added “The capping bilayer also enhances the perpendicular magnetic anisotropy of the magnetic layers.”

4. The authors have shown only one set of inputs (1 and 1) to demonstrate NAND and AND gate operation. However, they should demonstrate all four possible configurations (0 and 0) which need to result in 1 as output for demonstrating NAND gate operation and 0 for AND gate operation.

Response: Please refer to Fig. 5c, we do have demonstrated all the NAND and AND operations, including (1,1), (1,0), (0,1) and (0,0). The four dashed lines isolate the operations for the four sets of inputs and operations.

5. The authors need to proofread the manuscript thoroughly. There are a few typos and grammatical mistakes in the text.

Response: We have very carefully checked the typos in our manuscript, thanks for pointing that out.

Reviewer #2 (Remarks to the Author):

Comment 2-1: The authors present a novel type of DMI – intralayer one – as a missing element among the DMI family. However, intralayer DMI was introduced in 2019 in [Nat. Mater. 18, 679–684 (2019). <https://doi.org/10.1038/s41563-019-0386-4>] as an interaction, which favors one sense of rotation of spins in the same FM layer. In addition, a novel RKKY-type DMI has been found in synthetic ferromagnets [Nat Commun 12, 3280 (2021), <https://doi.org/10.1038/s41467-021-23586-y>; Nano Letters 2023 23 (18), 8690-8696, <https://doi.org/10.1021/acs.nanolett.3c02607>]. Recently, the DMI family has been widened by compositional gradient-induced DMI [Acta Materialia 241, (2023) 118383, <https://doi.org/10.1016/j.actamat.2022.118383>; Nano Lett. 2022, 22, 24, 10128–10133, <https://doi.org/10.1021/acs.nanolett.2c03973>].

Response: We thank the referee for his/her time spent on reviewing our manuscript. However, if the referee could read Nat. Mater. 18, 679–684 (2019) more carefully, he/she would agree that the “intralayer DMI” in that paper (see the captured figure below) actually referred to the conventional “short-range interfacial DMI” between adjacent atomic spins (see the classification in Fig. 1a in our manuscript) and conceptually distinct from the long-range intralayer DMI between neighboring domains which we for the first time propose and experimentally verify in our present manuscript (Fig. 1c in our manuscript). To avoid confusion, we use “long-range intralayer DMI” rather than simply “intralayer DMI” in our revised manuscript. The long-range intralayer DMI is more analogous to the long-range interlayer DMI in refs. 29-35 but occurs in the same magnetic layer.

We also do NOT believe the novelty of the long-range intralayer DMI in our present manuscript was affected by or directly related to any of the interlayer or bulk DMI effects in the synthetic ferromagnets and composition-gradient ferromagnet. In our revised manuscript, we have cited several more representative “RKKY DMI” references as the examples of a long-range DMI effects (such as Ref. 29-35, 61,62,72).

[Redacted]

Captured “intralayer” DMI schematic from Fig. 1a of Nat. Mater. 18, 679–684 (2019).

Comment 2-2: The idea presented in the manuscript looks interesting, but the experimental evidence and explanation are still puzzling. The three-side model used describes the interfacial DMI, where the interacting spins can have any direction. In the present paper the interacting spins are orthogonal because they are in the neighboring domains with different magnetization vectors; one is in-plane and another is out-of-plane. However, in a real system two domains are separated by a finite-width domain wall, but the model does not take it into account.

Response: Stimulated by the questions of the referee, in our revised manuscript we have added a domain wall between the coupling domains in Fig. 1c and Fig. 4a and a discussion that “**The long-range intralayer DMI describes the HM-mediated chiral coupling of neighboring magnetic domains separated by a magnetic domain wall within the same magnetic layer.**”

In our view, the Lert-Fert three-point model properly describes all the DMI effects, at least, Eq. 3 and Eq. 4 applies to the short-range interfacial DMI, long-range interlayer DMI, and the long-range intralayer DMI. The long-range intralayer DMI we report can be viewed as the lateral version of an interlayer DMI effect. Similar to the interlayer DMI and the interfacial DMI, the long-range intralayer DMI also maximizes in strength for orthogonal domains (see ref.33) and lowers when the configuration is tilted away from the perfect orthogonal configuration. In other word, the DMI strength is always given by the orthogonal components of the two coupling spins or domains ($\mathbf{M1} \times \mathbf{M2}$ or $\mathbf{s1} \times \mathbf{s2}$).

The short-range chiral coupling of a magnetic domain with the atomic spins of its domain wall has been described by the conventional interfacial DMI, which has been known for a decade to the cause of a **symmetric switching phase diagram** in Fig. 2c of our manuscript and the requirement of the in-plane magnetic field for deterministic switching of PMA samples via domain wall depinning and propagation. Please see Refs. 6,9,11 for more discussions.

Comment 2-3: In my opinion, the discovered effect originates from an interfacial DMI, which brings magnetization switching asymmetry.

Response: Honestly, it has been a long-standing consensus that the short-range interfacial DMI can ONLY cause a **symmetric switching phase diagram** shown in Fig. 2c of our manuscript (see Refs. 6,9,11). ONLY the intralayer DMI or interlayer DMI between neighboring orthogonal magnetic domains can cause an asymmetric switching. In absence of the interlayer DMI, the intralayer DMI is the only mechanism that can explain the striking phenomena we experimentally demonstrate from the various HM/FM heterostructures (Fig. 2b,d,f), such as strongly asymmetric switching, hysteresis loop shift in absence of in-plane direct current, and switching of perpendicular magnetization purely by an in-plane magnetic field.

In our revised manuscript, we have discussed that “**The long-range intralayer DMI in PMA HM/FM heterostructures manifests as an effective perpendicular magnetic field (H_{DMI}^z , Fig. 1c) that promote or hinder the switching of perpendicular magnetization, ultimately leading to a number of striking consequences, e.g., strong asymmetry in the switching density, hysteresis loop shift in absence of in-plane direct current, and switching of perpendicular magnetization purely by an in-plane magnetic field. None of these characteristics can be attributed to the short-range interfacial DMI.**”

Comment 2-4: To prove the intralayer origin, additional experiments are needed to be done. First, BLS measurements are crucial to probe spin wave dynamics and extract the DMI value and sign for each layered system under investigation. It is not correct to operate DMI values from literature because it is well-known that DMI is extremely sensitive to the crystal structure and quality of interfaces. Second, the precise structural properties including a cross-sectional view with subatomic resolution is required. It helps to estimate interfaces and the atomic structure. Third, magnetic properties have to be measured and collate with the structure including magnetically dead layers. It gives the effective M_s and anisotropy. The effect of DMI on magnetization switching cannot be discuss without whole understanding of all energy contributions.

In conclusion, I'd like to recommend conducting a comprehensive investigation in order to prove the existence of the proposed intralayer DMI, because now it seems just like a conventional interfacial DMI.

Response: First, as we have extensively addressed above and in the manuscript, **the conventional interfacial DMI has no chance to explain the experiments** (this is apparent if the referee is familiar with the difference between the interfacial and interlayer DMIs). We have unambiguously demonstrated the remarkable asymmetries and their characteristics in the current and field switching of the PMA samples, and we have also quantified the values of the out-of-plane effective field of the intralayer DMI. **No mechanism other than the long-range intralayer DMI can explain the strong switching asymmetries or the out-of-plane effective DMI field in the magnetic bilayers. These main observations and discussions cannot be altered regardless of the exact values of the interface sharpness, the DMI strengths, the saturation magnetization.**

Stimulated by the referee's comment, in our revised manuscript we have added **Extended Data Table 1** (saturation magnetization, anisotropy field, etc), Extended Data Fig. 1 (cross-sectional STEM image of the interface, thickness-dependent magnetic moment), and discussions on growth, interfaces and DMI of the samples:

Page 2: “As described in detail in the Sec. Method, these samples are sputter-deposited with our optimized growth protocol that typically yields reasonably sharp interfaces and no obvious intermixing or magnetic dead layer^{36,38-40} (e.g., for Pt/Co samples see the results of scanning transmission electron microscopy and the thickness-dependent magnetic moment measurements in the Extended Data Fig. 1). These samples exhibit high magnetization and strong interfacial PMA energy density ($M_s \approx 1200$ emu/cm³ for FeCoB and ≈ 1400 emu/cm³ for Co, $K_s \approx 1.1-2.0$ erg/cm², see Extended Data Table 1), which reaffirms the sharp interfaces of these samples since interfacial intermixing, if significant, would substantially degrade the apparent magnetization and the interfacial PMA^{39,40}.”

Page 7: “Here, the D_s values are adapted from **previous reports on corresponding magnetic interfaces that have similar sharpness and interfacial PMA energy density as our samples do in this study** (i.e., D_s is 0.21×10^{-7} erg/cm for Ir/FeCoB,⁴⁹ 0.38×10^{-7} erg/cm for W/FeCoB,⁵⁰ 0.22×10^{-7} erg/cm for Ta/FeCoB,⁵¹ 0.06×10^{-7} erg/cm for Ti/FeCoB,⁵² 1.2×10^{-7} erg/cm for Pt/Co,³⁸ 0.34×10^{-7} erg/cm for Ir/Co⁴⁹ and 0.01×10^{-7} erg/cm for Pd/Co⁵³, also see Extended Data Table 1). Note that quantification of the D_s values of these PMA samples from standard Brillouin light scattering (BLS) or loop shift measurements is prevented because the electromagnet of BLS setups available to us (≤ 2 kOe)^{38,54} cannot overcome the strong PMA field of our samples (up to 10 kOe, see Extended Data Table 1) to align the magnetization in-plane for the BLS analysis and because the strong dependence of the switching field on the in-plane field at zero dc current (Fig. 3b) invalidates the loop shift technique for these samples. For all the heterostructures, the current density asymmetry increases with the magnitude of D_s , while it reverses the sign for positive D_s compared to the negative D_s case. Such a direct correlation of the switching asymmetry to the sign and strength of the interfacial DMI further reaffirms that the long-range intralayer DMI is the mechanism of the observed switching asymmetry. **This conclusion is qualitatively robust and unaltered by any uncertainty of the used D_s values.**”

Page 11: “ **Sample fabrications:** Magnetic heterostructures of Ir 5.4/FeCoB 1, W 4/FeCoB 1, Ta 5/FeCoB 1, Cr 5/Ti 1/FeCoB 1, Pt 5/Co, Ir 5/Co 1, and Pd 5/Co 1 are sputter-deposited on oxidized Si substrates (the numbers are layer thicknesses in nanometer, FeCoB = Fe₆₀Co₂₀B₂₀). Each sample is seeded by a 1 nm Ta layer for improved the adhesion and the smoothness and protected from oxidation by a MgO 1.6/Ta 1.6 bilayer that is fully oxidized upon exposure to the atmosphere. The capping bilayer

also enhances the perpendicular magnetic anisotropy of the magnetic layers. Each layer was sputtered at a low rate (e.g., ≈ 0.007 nm/s for Co and FeCoB, ≈ 0.013 nm/s for Ir, ≈ 0.011 nm/s for W, ≈ 0.014 nm/s for Pt, ≈ 0.033 nm/s for Ta, ≈ 0.009 nm/s for Cr, ≈ 0.012 nm/s for Pd, ≈ 0.005 for Ti, and ≈ 0.004 nm/s for MgO) by introducing an oblique orientation of the target to the substrate and by using low magnetron sputtering power to minimize intermixing. No thermal annealing was performed on the samples. The layer thicknesses are estimated using the calibrated deposition rates and the deposition time and then verified by scanning transmission electron microscopy measurements. The base pressure during deposition is below 5×10^{-9} Torr. These samples are patterned into $5 \times 60 \mu\text{m}^2$ Hall bars by photolithography and ion milling, followed by deposition of Ti 5/Pt 150 as the electrical contacts for switching measurements.”

We note that the saturation magnetization and anisotropy field remain unchanged for a fixed device and do not affect the analysis of the qualitative characteristics of the switching experiments. Technically, a BLS experiment requires the magnetization to be aligned fully in-plane and is thus usually performed only on in-plane magnetized samples or very poor PMA samples that can be aligned to the film plane by an in-plane magnetic field of the BLS setup. However, the samples in our present study have strong perpendicular magnetic anisotropy fields that are much greater than the maximum in-plane field of the BLS setups we can have access in collaboration. In this case, we think it is helpful to refer to the literature values of the DMI strength of corresponding **interfaces** that are **sharp and similar interfacial perpendicular magnetic anisotropy** because they are typically close enough for a semi-quantitative investigation of the correlation of the experimentally observed DMI field and the interfacial DMI strength. We do agree that DMI may vary if the interfaces are substantially degraded such that the interfacial SOC is reduced, which is, however, not the case for our samples.

Minor remarks

1. Kerr microscope images with complete in-plane saturation magnetization are needed as an addition to Fig.4b.
2. A detailed description of the sample fabrication process is required.

Response: Following the referee’ suggestions, we have added a Kerr image of the complete in-plane magnetization state in Fig. 4f and a detailed description of the sample preparation in the Method section of our revised manuscript:

Page 11: “**Sample fabrications:** Magnetic heterostructures of Ir 5.4/FeCoB 1, W 4/FeCoB 1, Ta 5/FeCoB 1, Cr 5/Ti 1/FeCoB 1, Pt 5/Co, Ir 5/Co 1, and Pd 5/Co 1 are sputter-deposited on oxidized Si substrates (the numbers are layer thicknesses in nanometer, FeCoB = $\text{Fe}_{60}\text{Co}_{20}\text{B}_{20}$). Each sample is seeded by a 1 nm Ta layer for improved the adhesion and the smoothness and protected from oxidization by a MgO 1.6 /Ta 1.6 bilayer that is fully oxidized upon exposure to the atmosphere. The capping bilayer also enhances the perpendicular magnetic anisotropy of the magnetic layers. Each layer was sputter-deposited at a low rate (e.g., ≈ 0.007 nm/s for Co and FeCoB, ≈ 0.013 nm/s for Ir, ≈ 0.011 nm/s for W, ≈ 0.014 nm/s for Pt, ≈ 0.033 nm/s for Ta, ≈ 0.009 nm/s for Cr, ≈ 0.012 nm/s for Pd, ≈ 0.005 for Ti, and ≈ 0.004 nm/s for MgO) by introducing an oblique orientation of the target to the substrate and by using low magnetron sputtering power to minimize intermixing. No thermal annealing was performed on the samples. The layer thicknesses are estimated using the calibrated deposition rates and the deposition time and then verified by scanning transmission electron microscopy measurements. The base pressure during deposition is below 5×10^{-9} Torr. These samples are patterned into $5 \times 60 \mu\text{m}^2$ Hall bars by photolithography and ion milling, followed by deposition of Ti 5/Pt 150 as the electrical contacts for switching measurements.”

Reviewer #3 (Remarks to the Author):

The authors report the discovery of the "intralayer Dzyaloshinskii-Moriya interaction (DMI) effect". They argued that the intralayer DMI is distinct from the already-known DMI inside a magnetic layer (also intralayer DMI). The "new" intralayer DMI refers to the DMI interaction among adjacent magnetic domains with different anisotropies that favor either in-plane or out-of-plane alignment. The manuscript presents current-induced switching experiments of perpendicular magnetization in different kinds of samples with interfacial DMI and an inhomogeneous perpendicular magnetic anisotropy. Due to the following reasons, the manuscript is unsuitable for Nature Communications.

The polar MOKE measurements reveal that there is a magnetic anisotropy non-uniformity in one of the samples (W/CoFeB), with some domains with low perpendicular anisotropy and others with large perpendicular anisotropy. This is a central issue in the manuscript since the proposed intralayer DMI relies on the anisotropy non-uniformity. However, there needs to be more information about the sizes of the domains and the domain walls that separate them. This is important since the interfacial DMI is a type of exchange interaction among neighboring magnetic moments, with a spatial extension of some nanometers, which is relatively small compared to the domain wall size. Then, I am trying to understand why the authors assume an interfacial DMI considering each magnetic domain as a single spin (see Eq. 3), which means the interfacial DMI is active on a hundred nanometers scale. This is an extreme and unjustified assumption that makes the interpretation of the results very hard to believe.

Response: The referee raised two important questions: (1) whether the requirement of anisotropy non-uniformity does not apply to real samples and thus a "central issue" and (2) whether the long-range chiral coupling of two neighboring orthogonal domains is possible in the presence of a magnetic domain wall. We carefully address the two questions below.

As far as we understand, anisotropy variations occur commonly in realistic PMA samples, regardless the growth technique (sputtering, molecular-beam epitaxy, or others). This is indicated by the gradual, memristor-like, or even partial switching behaviors of PMA samples (see the figures below), which is in contrast to the perfectly sharp two-state switching expected for a uniform PMA sample (only $\pm M_z$ states). Strong two-magnon scattering, arising from anisotropy variations have also widely observed for in-plane samples. Therefore, **anisotropy variation is the common rather than specific case and our intralayer DMI model taking into account the anisotropy distributions is a realistic, accurate description of the samples.** We also emphasize that ONLY the intralayer DMI between neighboring magnetic domains explains the strongly asymmetric switching phenomena of the a number of PMA samples in Fig. 2b,d,f, which are unexpected for interfacial DMI model describing the short-range coupling of neighboring atomic spins of a uniform PMA sample. The switching phase diagram should be strictly symmetric in terms of switching current density and the in-plane bias field, please refer to Fig. 2c of our revised manuscript.

[Redacted]

Remarkable memristor-like gradual Current switching of a PMA Co/Ni multilayer sample, showing remarkable dependence of the switching ratio on the highest current density, Captured from Nat. Mater.15, 535–541 (2016).

Current switching of a PMA FeCoB sample, showing remarkable dependence of the switching ratio on the in-plane field, Captured from Nat. Commun. 13, 4447 (2022)

In our revised manuscript, we have added these discussions:

Page 2: “**The long-range intralayer DMI in PMA HM/FM heterostructures manifests as an effective perpendicular magnetic field (H_{DMI}^z , Fig. 1c) that promote or hinder the switching of perpendicular magnetization, ultimately leading to a number of striking consequences, e.g., strong asymmetry in the switching density, hysteresis loop shift in the absence of in-plane direct current, and switching of perpendicular magnetization purely by an in-plane magnetic field. **None of these characteristics can be attributed to the short-range interfacial DMI.****”

Page 4: “**The switching current densities after subtraction of the shift, $j_{\uparrow 0} = j_{\uparrow} - j_{\text{shift}}$ and $j_{\downarrow 0} = j_{\downarrow} - j_{\text{shift}}$, decrease as $|H_x|$ increases and remain symmetric for $\pm H_x$, in agreement with the expectation of anti-damping torque-driven depinning of **chiral domain walls with short-range interfacial DMI.****”

Page 5: “As schematically shown in Fig. 4a, a realistic magnetic layer typically has non-uniformity in its magnetic anisotropy, evidence of which includes the **widely existing two-magnon scattering damping** of in-plane magnetized HM/FM bilayers^{44,45} and **gradual, memristor-like, or even partial**

electrical switching behaviors of PMA samples^{46,47} (see more examples in Ref. 9 and references therein). Note that a perfectly uniform PMA sample should only have a sharp two-state switching.”

To address the second question “whether the long-range chiral coupling of two neighboring orthogonal domains is possible in the presence of a magnetic domain wall”, in our revised manuscript we have added the domain wall width of our PMA samples in Extended Data Table 1 and an independent paragraph on validity of the coupling length of the intralayer DMI:

Page 7: “**We also note that the occurrence of the long-range intralayer DMI is interesting but not too surprising in terms of the chiral coupling distance. Using the domain wall width relation⁵⁵ $\Delta = [A/(H_k M_s/2 + N_y 2\pi M_s^2)]^{1/2}$ with the exchange stiffness A of ≈ 1.0 $\mu\text{erg/cm}$ for Co ⁵⁶⁻⁵⁸ and ≈ 0.8 $\mu\text{erg/cm}$ for FeCoB ^{59,60} and the demagnetizing factor across the wall⁵⁵ $N_y \approx (1 + \Delta/t_{\text{FM}})^{-1}$, we estimate the width of the magnetic domain wall (Δ) that separates the domain walls coupled by the HM-mediated intralayer DMI layer is 3-6 nanometers for the samples in this study (Extended Data Table 1), which is within the typical range of the DMI effects.^{32,35,48,61,62} For example, significant interlayer DMI coupling has been reported between neighboring orthogonal domains separated by a heavy metal layer that is typically several nm thick.^{32,35,61,62} A long-distance chiral coupling is also suggested in an early experiment on the chiral coupling of two perpendicular domains via a deliberately fabricated in-plane magnetic domain of up to 200 nm long (which was reported before the discovery of the long-range DMIs and attributed to the short-range interfacial DMI)⁴⁸.”**

As to the size of the in-plane domains, it certainly varies depending on the magnetization states, please refer to the MOKE images.

We believe the effects of the long-range interlayer DMI, chiral coupling of nanomagnets, and very strong asymmetry in our manuscript do suggest that the DMI effect can be significant even in a large length scale. The striking contrast between the length scales of the long-range intralayer DMI of neighboring magnetic domains and the short-range interfacial DMI of adjacent atomic spins signify that the intralayer DMI is a distinct, new, fundamental effect that warrants a timely publication in a high-profile journal like Nature Communications. We hope these revisions have addressed the referees’ stimulating questions.

REVIEWER COMMENTS

Reviewer #1 (Remarks to the Author):

The authors have carefully considered my comments and addressed most of the points.

Reviewer #2 (Remarks to the Author):

The authors have made major revision to their manuscript based on the comments and suggestions. The revision has adequately addressed all the comments and has made the manuscript more compelling. The paper is recommended to be published as is.

Reviewer #3 (Remarks to the Author):

I regret to report that I am not persuaded by the author's response. First, the experiments do not show domain walls as small as those estimated with the Mougine et al. formula, which does not consider DMI but, according to the authors, gives a domain wall typical of systems with DMI. Second, there is no justification for treating the domains as macrospins and ignoring the gradual variation of magnetization. It is not enough to add a schematic domain wall in some figures and include some numbers. The existence of the DMI interaction on a length scale greater than usual is also not justified. All these unjustified assumptions weaken the explanation of the experiments based on the long-range DMI.

Response letter

Reviewer #1:

The authors have carefully considered my comments and addressed most of the points.

Reviewer #2:

The authors have made major revision to their manuscript based on the comments and suggestions. The revision has adequately addressed all the comments and has made the manuscript more compelling. The paper is recommended to be published as is.

Response: We are grateful to the Reviewer #1 and reviewer #2 for their recommendation for publication.

Reviewer #3:

Comment 3-1: I regret to report that I am not persuaded by the author's response. First, the experiments do not show domain walls as small as those estimated with the Mougine et al. formula, which does not consider DMI but, according to the authors, gives a domain wall typical of systems with DMI.

Response: First of all, in sharp contrast to the referee's speculation, without any supporting references or specified mechanisms, that the domain width of heavy metal/ferromagnet (HM/FM) bilayers with strong PMA should be "a hundred nanometers"(see the first report of the Reviewer #3), **small domains widths (Δ) of a few nanometers are very commonly estimated for PMA samples with DMI in the spintronic community including the leading domain wall groups using the same formula as Equation (6) in our present manuscript.**

In our revised manuscript, we have added that "**Theories and simulations have also indicated that Δ in ultrathin FMs is essentially independent of the DMI strength⁶⁰ and the domain wall configuration (Bloch or Néel) ⁶¹. Therefore, Equation (6) has been widely applied to various PMA heterostructures with DMI.⁵⁶⁻⁵⁹ Note that the simplified relation $\Delta_{\text{upper}} \approx \sqrt{2A/H_k M_s}$ in the literature⁶² has ignored the domain wall shape anisotropy and only yields the upper limit of the domain wall width.**" The original data in Ref. 60 (see $d=1$ nm data, G.S.D. Beach group, PRB 95, 174423) and Ref. 61 (Christopher Marrows group, PRL 127, 127203 (2021)) are attached below.

Ref. 60 (see $d=1$ nm data, PRB 95, 174423)

Ref. 61 (PRL 127, 127203 (2021))

[Redacted]

[Redacted]

We also note that the theoretical values of Δ in above two calculations are a few nm in most cases. For the convenience of the editors and the referees, we provide a few more examples of small domain wall width in heavy metal/ferromagnet heterostructures below:

Example 1: Small Δ of 3.72 nm has been estimated by **Albert Fert group** in Sci. Adv. 4, eaat0415 (2018).

[Redacted]

Example 2: $\Delta = 4.3$ nm has been estimated by **Stuart S. P. Parkin group** in Nat. Commun. 5, 3910 (2014)

[Redacted]

Example 3: Δ of 5.5-8.6 nm has been estimated by **CNRS/Spintec** in PRL 99,217208 (2007).

[Redacted]

We note experimental quantification of a domain wall width of below 10 nm (3-6 nm in this work), if not impossible, is extremely challenging and far beyond the scope of our present work, according to our careful literature investigation and discussions with several leading experts on magnetic microscopies (e.g., state-of-art Lorentz-STEM, XMLD-PEEM, spin-polarized LEEM, spin-polarized STM, etc.). The challenges include but not limited to:

- (1) The very weak magnetic signals of the domain walls of the ultrathin Co and FeCoB layers (only 1 nm thick) in this study are extremely challenging to detect by any microscopic tools in the world;
- (2) Due to the limitation of magnetically spatial resolution, there has NO report of a technique that had reliably identified a magnetic domain wall width of below 10 nm;
- (3) There appears to be NO high-resolution microscopic instruments, such as Lorentz-scanning transmission electron microscopy (L-STEM), that can apply an in-plane magnetic field of as strong as 1-3 kOe to form a domain wall between the perpendicular and in-plane domains (we are aware that state-of-art L-STEM can have a perpendicular magnetic field that can be tilt by only a few degrees, **NOT the required 90 deg**, relative to the sample normal.)
- (4) The HM/Co or FeCoB/MgO/Ta samples with interfacial PMA and capping layer do not allow surface-sensitive measurements such as spin-polarized STM or spin-polarized LEEM.

To avoid similar questions from future audiences, we have added the above points in our revised manuscript:

“we estimate the width of the magnetic domain wall that separates the domains coupled by the HM-mediated intralayer DMI layer is 3-6 nanometers for the samples in this study (Extended Data Table 1). **These Δ values agree well with those estimated for typical PMA HM/FM heterostructures in the literature reports^{55,56,59-62} and are within the typical range of the DMI effects.^{32,35,48,68,69} ...Note that direct experimental quantification of such few-nm magnetic domain width between the in-plane and perpendicular domains of strong PMA samples has been extremely challenging and beyond the scope of this work.** To the best of our knowledge, there has been no report of a microscopic technique that simultaneously had a sensitivity capable of the very weak magnetic signal of the narrow domain walls of only 1 nm thick magnetic layer, a magnetically spatial resolution of ~1 nm or below, and an in-plane magnetic field of 1-3 kOe to form a magnetic domain wall between adjacent in-plane and perpendicular domains within strong PMA samples.”

Comment 3-2: Second, there is no justification for treating the domains as macrospins and ignoring the gradual variation of magnetization. It is not enough to add a schematic domain wall in some figures and include some numbers.

Response: Actually, we had already addressed this question in Comment 2-2 of our previous response letter. Our treatment of the DMI field exerted on the perpendicular domain by the in-plane domain is justified simply because two parallel magnetic moments always have no DMI coupling, within the frame of any DMI effects. When a low-anisotropy domain (\vec{M}) is tilted away from the perpendicular direction, only its in-plane component ($\vec{M}_{\text{in-plane}}$) can have DMI interaction with the adjacent perpendicular domain (\vec{M}_2), while its perpendicular component ($\vec{M}_{\text{perpendicular}}$) does not, i.e., $(\vec{M}_{\text{in-plane}} + \vec{M}_{\text{perpendicular}}) \times \vec{M}_2 \equiv \vec{M}_{\text{in-plane}} \times \vec{M}_2$ and $\vec{M}_{\text{perpendicular}} \times \vec{M}_2 \equiv 0$.

In our revised manuscript, we have added that “Here, we only consider the DMI coupling between the perpendicular \vec{M}_2 and the magnetic component that is orthogonal to \vec{M}_2 (noted as in-plane \vec{M}_1) because any perpendicular component ($\vec{M}_{\text{perpendicular}}$) of a tilted domain or domain wall will have no DMI coupling with \vec{M}_2 for any DMI effects (i.e., $\vec{M}_{\text{perpendicular}} \times \vec{M}_2 = 0$). **Apparently, the collection of the in-plane component of a magnetic domain, which is the only part at work for the intralayer DMI effect in this study, is effectively an in-plane macrospin for the perpendicular domain \vec{M}_2 .**”

Comment 3-2: The existence of the DMI interaction on a length scale greater than usual is also not justified. All these unjustified assumptions weaken the explanation of the experiments based on the long-range DMI.

Response: We strongly disagree with the assertion of the referee that the 3-6 nm length scale of our long-range intralayer DMI is **greater than usual**.

(1) First, this assertion is conflicted with the comments of the same referee in his/her last report that **even the short-range interfacial DMI**, which describes the chiral coupling of two adjacent atomic spins, had “**a spatial extension of some nanometers**”. We do not understand why the referee attempted to claim a long coupling distance for a short-range interfacial DMI effect but attempted to preventing the publication of our manuscript by speculating that a similar coupling distance was, however, unjustified for any long-range DMI effects.

(2) In our view, **it has been well established that the maximum coupling distance of a long-range DMI effect can be at least several nanometers**. For example, significant DMI coupling has been commonly reported for orthogonal ferromagnetic layers separated by a non-magnetic **heavy metal** layer of several nanometers in thickness. For the convenience of the editors and the referees, we list a few examples below:

samples	Maximum DMI coupling distance (nm)	Measured DMI field H_{DMI} (Oe)	Ref.
Co/[Pt/Ru/Pt](3.7)/Co	> 3.7	14	Nat. Mater. 18, 703 (2019)
Co ₇₅ Si ₁₅ B ₁₀ /Pt(2.2)/Co ₇₅ Si ₁₅ B ₁₀	> 2.2	600	
CoFeB/W(4.5)/CoFeB	> 4.5	2	ACS Nano17, 9049 (2023)
Co/[Pt/Ru/Pt](3.1)/Co	> 3.1	310	Adv. Funct. Mater. 33, 2301731 (2023)
Co/[Pt/Ir/Pt](2.7)/Co	> 2.7	14	Nano Lett. 22, 6857 (2022)
CoFeB/Pt(2.5)/Co	> 2.5	37	Phys. Rev. Applied 18, 034046 (2022)
Co(1)/[Pt/Ir/Pt](2.7)/Co(0.9)	> 2.7	15	Nano Lett. 23, 8690 (2023)

In our view, we had clearly discussed in our manuscript that the length scale of our intralayer DMI in our present work is within the typical range of the long-range DMI effects:

“we estimate the width of the magnetic domain wall (Δ) that separates the domain walls coupled by the HM-mediated intralayer DMI layer is 3-6 nanometers for the samples in this study (Extended Data Table 1). **These Δ values agree well with those estimated for typical PMA HM/FM heterostructures in the literature reports^{55,56,59-62} and are within the typical range of the DMI effects.^{32,35,48,68,69} For example, significant interlayer DMI coupling has been reported between neighboring orthogonal domains separated by a heavy metal layer that is typically several nm thick.^{32,35,68,69}** A long-distance chiral coupling is also suggested in an early experiment on the chiral coupling of two perpendicular domains via a deliberately fabricated in-plane

magnetic domain of up to 200 nm long (which was reported before the discovery of the long-range DMIs and attributed to the short-range interfacial DMI)⁴⁸.”

(3) Third, NO mechanism other than the long-range intralayer DMI can explain the strong switching asymmetries or the out-of-plane effective DMI field in the magnetic bilayers. In physics, the long-range intralayer DMI mechanism is required to explain the experimental observation of strongly asymmetric switching.

In summary, we believe we have adequately addressed all the questions of the referees and we have clearly demonstrated the effects of the long-range interlayer DMI, chiral coupling of nanomagnets, and very strong asymmetry in our manuscript. These experimental results unambiguously indicate that the DMI effect can be significant even in a large length scale. The striking contrast between the length scales of the long-range intralayer DMI of neighboring magnetic domains and the short-range interfacial DMI of adjacent atomic spins signify that the intralayer DMI is a distinct, new, fundamental effect that warrants a timely publication in a high-profile journal like Nature Communications.

REVIEWER COMMENTS

Reviewer #3 (Remarks to the Author):

I recognize the authors' efforts; they have significantly modified the manuscript since its first version. However, some issues must be amended before publication.

Comment 3-1:

The authors have given enough references about the small domain walls, which I was unaware of. I understand that measuring these walls is beyond the scope of the manuscript. There is no problem here.

Comment 3-2:

The explanation of this point in the new version is correct for M1 and M2 according to Fig. 4a, where the DM field points to $-z$ if $D > 0$ and $H_x > 0$. However, it is not evident that the TOTAL DM field exerted over an entire perpendicular domain point in the $-z$ direction. For example, if we consider a third magnetic moment (M3) parallel to M1 due to $H_x > 0$ and located on the right side of the perpendicular domain M2, the DM field that M3 would produce on M2 would be opposite to the field of M1 on M2, due to the change of sign of the vector D_{13} respect to D_{12} . Then, the total DMI field exerted on the perpendicular domain could point at $-z$ or $+z$, depending on the shape of the perpendicular domain. Then, the origin of the DMI field seems more complex, and this simple reasoning raises doubts about the interpretation used by the authors. Also, the DM energy between M1 and M2, for the case of Fig. 4a ($D > 0$, $H_x > 0$), is maximum since the DM vector D_{12} and the cross product between M1 and M2 are parallel, which may be an indication of the minimal effect of the intralayer DMI and must be discussed.

Comment 3-3:

Regarding the length scale of the DMI, the usual scale is only a few neighbors, i.e., 1 or 2 nm. The point is that intralayer DMI should occur only between the spins at the domain wall boundaries and not between spins further away within the in-plane and out-of-plane domains, which should be mentioned. The Levy-Fert model tells us that the magnitude of the DM vector decays with the distance between spins, which can be easily estimated to have a more realistic idea of the DM field used to explain the experiments.

Concerning the Levy-Fert (not "Lert-Fert") model, references 3, 21, and 48 are cited on page 6. However, the appropriate references are P. M. Levy and A. Fert, Phys. Rev. B 23, 4667 (1981), where the theory was developed, and Ref. 23.

Response letter

Reviewer #3:

I recognize the authors' efforts; they have significantly modified the manuscript since its first version. However, some issues must be amended before publication.

Comment 3-1:

The authors have given enough references about the small domain walls, which I was unaware of. I understand that measuring these walls is beyond the scope of the manuscript. There is no problem here.

Response: We thank the referee for agreeing that we had addressed all the previous comments of the referee. We address the two new questions of the referee below.

Comment 3-2:

The explanation of this point in the new version is correct for M1 and M2 according to Fig. 4a, where the DM field points to $-z$ if $D>0$ and $H_x>0$. However, it is not evident that the TOTAL DM field exerted over an entire perpendicular domain point in the $-z$ direction. For example, if we consider a third magnetic moment (M3) parallel to M1 due to $H_x>0$ and located on the right side of the perpendicular domain M2, the DM field that M3 would produce on M2 would be opposite to the field of M1 on M2, due to the change of sign of the vector D_{13} respect to D_{12} . Then, the total DMI field exerted on the perpendicular domain could point at $-z$ or $+z$, depending on the shape of the perpendicular domain. Then, the origin of the DMI field seems more complex, and this simple reasoning raises doubts about the interpretation used by the authors. Also, the DM energy between M1 and M2, for the case of Fig. 4a ($D>0$, $H_x>0$), is maximum since the DM vector D_{12} and the cross product between M1 and M2 are parallel, which may be an indication of the minimal effect of the intralayer DMI and must be discussed.

Response: We thank the referee for the interesting question on the non-cancelling total DMI. The total DMI would be zero when a perpendicular domain (spin) interacts with the parallel right and left in-plane domains (spins), which appears to be a question not only to the intralayer DMI but also to the interfacial DMI and interlayer DMI. As shown in the figure below, the various DMI effects, including the interfacial DMI, interlayer DMI, and intralayer DMI, appear to cancel out for the center spin (or domain) when the DMI vectors are of opposite signs on the left and right sides (1 and 2 in the figure). However, the nonzero total DMI effects in practical samples are proved by the consequences of various DMI effects (e.g., interfacial DMI-induced frequency difference between counterpropagating Damon-Eshbach spin waves, interlayer and intralayer DMI-induced switching asymmetry). This likely suggests that the total DMI effects of practical samples are mainly from the spins (domains) and heavy metal atoms that are in microscopic configurations more complex than those plotted in the figure. Modeling the non-zero DMI effects in the presence of multiple domains is worth future efforts but is beyond the scope of our present manuscript. We would appreciate it very much if the referee had a better idea about such modeling.

Total DMI. **a**, interfacial DMI, **b**, interlayer DMI, **c**, intralayer DMI. FM and HM are short for the ferromagnetic layer and the heavy metal, respectively.

In our revised manuscript, we have added the above discussions in the Extended Data Note section of our revised manuscript (at the end of the manuscript document) and provided a toy model in which the intralayer DMI is nonzero when the moments of the spins (domains) are not coplanar with the heavy metal atoms. In the main text of our revised manuscript, we have also added that “**While the effects of the dimensions and relative location of the interacting domains on the magnitude of H_{DMI}^z still lacks a simple, unified analytical calculation due to the involvement of multiple complex microscopic configurations of the practical samples (see more discussions in the Extended Data Note 1), the nonzero total DMI effects in practical samples are proved by the consequences of various DMI effects (e.g., the interfacial DMI-induced frequency difference between counterpropagating Damon-Eshbach spin waves,^{49,50} the interlayer³⁰⁻³⁵ or intralayer DMI-induced switching asymmetry).**”

We also agree that the DMI energy is maximum when $\vec{D}_{12} // \vec{M}_1 \times \vec{M}_2$ such that the DMI field prefers to switch the perpendicular moment to lower the system energy. However, the DMI energy itself cannot directly switch the perpendicular domain because the status of a magnetic system is determined collectively by all the energy terms, not by the DMI energy. We do not consider this a point indicating any paradox or any typical question of other readers.

Comment 3-3:

Regarding the length scale of the DMI, the usual scale is only a few neighbors, i.e., 1 or 2 nm. The point is that intralayer DMI should occur only between the spins at the domain wall boundaries and not between spins further away within the in-plane and out-of-plane domains, which should be mentioned. The Levy-Fert model tells us that the magnitude of the DM vector decays with the distance between spins, which can be easily estimated to have a more realistic idea of the DM field used to explain the experiments.

Concerning the Levy-Fert (not "Lert-Fert") model, references 3, 21, and 48 are cited on page 6. However, the appropriate references are P. M. Levy and A. Fert, Phys. Rev. B 23, 4667 (1981), where the theory was developed, and Ref. 23.

Response: We thank the referee for the suggestions. On page 6 of our revised manuscript, we have added that “**the DMI vector $\vec{D}_{12} = \zeta D \hat{r}_1 \times \hat{r}_2$. Here, $\hat{r}_{1,2}$ is the unit vector linking the mediate HM atom and interacting moment $\vec{M}_{1,2}$, D is the interfacial DMI constant of the HM/FM interface, ζ is a parameter related to the dimensions of the interacting domains since the intralayer DMI is expected to be the strongest between the spins near the domain wall boundaries and decays between spins further away within the in-plane and out-of-plane domains.**” Again, we emphasize that the experiments of the significant long-range DMI effects, including the interlayer DMI mediated by several nm heavy metals have indicated an interaction length of several nanometers

is rather possible for long-range DMI effects. We hope the referee can be more tolerant and take the long-range interlayer and intralayer DMI effects into consideration when asserting the scale of the DMI.

We have corrected the typo of “Levy” and cited Ref.23 and Phys. Rev. B 23, 4667 (1981) for the Levy-Fert model. We also cited ref.48 for its employment of the Levy-Fert model for the chiral interaction of magnetic domains.

We believe that these additional revisions render the manuscript acceptable now to be published in *Nature Communications*.

REVIEWERS' COMMENTS

Reviewer #3 (Remarks to the Author):

The authors have discussed the raised points and improved the manuscript, which can now be published.